# Teaching via Best-Case Counterexamples in the Learning-with-Equivalence-Queries Paradigm

**Akash Kumar**[1,2]
akk002@ucsd.edu

**Yuxin Chen**[3]
chenyuxin@uchicago.edu

**Adish Singla**[1]
adishs@mpi-sws.org

[1]Max Planck Institute for Software Systems (MPI-SWS),
[2]UC San Diego,
[3]University of Chicago

## Abstract

We study the sample complexity of teaching, termed as "teaching dimension" (TD) in the literature, for the learning-with-equivalence-queries (LwEQ) paradigm. More concretely, we consider a learner who asks equivalence queries (i.e., "is the queried hypothesis the target hypothesis?"), and a teacher responds either "yes" or "no" along with a counterexample to the queried hypothesis. This learning paradigm has been extensively studied when the learner receives worst-case or random counterexamples; in this paper, we consider the optimal teacher who picks best-case counterexamples to teach the target hypothesis within a hypothesis class. For this optimal teacher, we introduce LwEQ-TD, a notion of TD capturing the teaching complexity (i.e., the number of queries made) in this paradigm. We show that a significant reduction in queries can be achieved with best-case counterexamples, in contrast to worst-case or random counterexamples, for different hypothesis classes. Furthermore, we establish new connections of LwEQ-TD to the well-studied notions of TD in the learning-from-samples paradigm.

## 1 Introduction

Learning-with-queries paradigm involves a learner who asks structured queries to a teacher in order to locate a target hypothesis. This paradigm has been extensively studied in machine learning and formal methods research, including automata learning [1, 2, 3, 4], model checking [5], oracle-guided synthesis (OGIS) [6], model learning [7], among others. Classical literature involves different kinds of queries that a learner could ask [1, 8, 9], such as *membership* queries (i.e., "is the queried instance consistent with the target hypothesis?") and *equivalence* queries (i.e., "is the queried hypothesis the target hypothesis?"). In this paper, we consider the learning-with-equivalence-queries (LwEQ) paradigm where the learner is only asking equivalence queries and the teacher responds to a query either "yes" or "no" along with a counterexample on which the current hypothesis disagrees with the target hypothesis. LwEQ paradigm captures a variety of important problem settings, such as counterexample-guided synthesis (CEGIS) [10], data augmentation (CEGAR) [11], and learning regular languages from counterexamples [12]. The focus of our work is to understand the query complexity for the LwEQ paradigm, i.e., the number of equivalence queries needed by the learner for the exact identification of a target hypothesis [9].

The query complexity clearly depends on the learner model (i.e., the query function deciding the next hypothesis to query), as well as the informativeness of the counterexamples provided by the teacher. In the literature, the query complexity for the LwEQ paradigm has been extensively studied when the learner receives worst-case or random counterexamples [1, 9, 13, 14, 15]. To show worst-case bounds [1, 13], classical works have studied a teacher who responds with worst-case counterexamples

to maximize the number of equivalence queries asked by the learner. Recent work [14] has studied this query complexity when facing a more benign teacher who provides counterexamples selected at random from a known probability distribution. In particular, [14] proposed a learning algorithm (i.e., a query function for the learner) that achieves $\mathcal{O}\left(\log n\right)$ query complexity on the expected number of random counterexamples for any hypothesis class of size $n$. When contrasting this bound with worst-case bounds, it shows that random counterexamples could lead to exponential improvement in the query complexity when compared with worst-case counterexamples [13, 14]. Along these lines, an important research question is to understand the query complexity where the counterexamples are picked by a more informed and helpful teacher, instead of random or worst-case counterexamples.

In this paper, we consider a more powerful teaching setting, where the optimal teacher picks best-case counterexamples to steer the learner towards a target hypothesis. Our goal is to characterize the query complexity for this optimal teacher (also, referred to as *teaching complexity* in the paper), for the LwEQ paradigm. This teaching complexity has been extensively studied for binary classification in the learning-from-samples (LfS) paradigm [16, 17] and is termed as "teaching dimension" (TD) [18, 19, 20, 21, 22, 23, 24, 25, 26, 27, 28, 29]. Beyond the LfS paradigm, various notions of teaching complexity have also been studied in other learning paradigms (see Section 1.1 for further details). We introduce a new notion of teaching complexity for the LwEQ paradigm, namely "LwEQ teaching dimension" (LwEQ-TD), capturing the number of queries needed by the learner when best-case counterexamples are provided by the optimal teacher. Our study of LwEQ-TD for some prominent hypothesis classes reveals the power of teaching via best-case counterexamples—we show a significant reduction in query complexity when compared to that for worst-case and random counterexamples. Furthermore, we establish several new connections of LwEQ-TD to the existing notions of TD in the LfS paradigm.[1] Table 1 provides a summary of these different teaching settings and learning paradigms; our main results and contributions are summarized below:

I. We characterize the query complexity for the optimal teacher in the LwEQ paradigm, termed as *learning-with-equivalence-queries teaching dimension* (LwEQ-TD). (see Section 3)

II. We study the query complexity in the LwEQ paradigm under different teaching settings: worst-case, random, and best-case, distinguished by the informativeness of counterexamples. We showcase the power of best-case counterexamples picked by the optimal teacher, in contrast to worst-case or random counterexamples, for different hypothesis classes, including Axes-aligned hyperplanes, Monotone monomials, and Orthogonal rectangles. (see Section 4)

III. We establish new connections between LwEQ-TD and LfS-TD by studying LwEQ-TD for different learner models based on the richness of their query functions. We show that LwEQ-TD is the same as wc-TD [18], RTD [22, 24], and NCTD [27] for a hypothesis class when restricting query functions to specific families. In general, LwEQ-TD is weaker than LfS-TD, e.g., LwEQ-TD is lower-bounded by local-PBTD [26, 29] of the hypothesis class when the learner's next query depends on the previous query. (see Section 5)

| Teaching / Learning | Worst-case Teacher | Random-case Teacher | Best-case Teacher |
|---|---|---|---|
| Learning-with-equivalence-queries (LwEQ) | Worst-case counterexamples [1, 30, 9, 31, 13] | Random counterexamples [14, 32] | LwEQ-TD **This work** |
| Learning-from-samples (LfS) | Worst-case examples (i.e., least informative) | i.i.d learning [16, 17] | LfS-TD / classical TD [18, 22, 25, 27, 29] |

Table 1: An overview of different teaching settings in the context of LwEQ and LfS paradigms.

## 1.1 Background and Related Work

**Learning-with-queries paradigm and equivalence queries.** Learning-with-queries paradigm was introduced in [1] which proposed $L^*$ algorithm for exact identification of DFAs (deterministic finite automaton) when the learner is allowed to ask membership queries and equivalence queries. Classical work has studied different types of queries (subset, membership, equivalence, correction, among others) [1, 8, 9]. Among these works, learning with membership queries has been explored in a variety of problem settings, such as PAC learning [16], active learning [33, 34, 35], and agnostic

---

[1]We will collectively refer to these different notions of TD in the LfS paradigm as LfS-TD.

learning [36]. In the learning-with-equivalence-queries (LwEQ) paradigm, the learner can ask only equivalence queries; furthermore, in our work, we consider *proper* queries, i.e., the queried hypothesis is within the hypothesis class (see Section 6 for a discussion on *improper* queries). In this LwEQ paradigm, [13] studied query complexity when the teacher picks worst-case counterexamples for some key hypothesis classes (e.g., DFA, NFA, Context-free Grammars), and showed an exponential lower bound. [14] studied random counterexamples, and proposed a learning algorithm, namely Max-Min, which achieves a substantially improved bound on query complexity. We will investigate the query complexity of Max-Min learner for the hypothesis class of Axes-aligned hyperplanes in Section 4.

**Teaching in the LfS paradigm for binary classification.** Algorithmic machine teaching, first introduced by [18, 37], studies the interaction between a teacher and a learner where the teacher's goal is to find an optimal sequence of training samples to teach a target hypothesis. [18] introduced a measure of teaching complexity, named teaching dimension (TD) of the hypothesis class, in the learning-from-samples (LfS) paradigm. The classical notion of TD in [18] characterized the minimum number of samples (i.e., examples) needed to teach a target hypothesis to a version-space learner who picks hypothesis within the version space *arbitrarily* (in an adversarial way). In the past two decades, several new teaching settings have been studied, driven by the motivation to lower teaching complexity and to find settings for which TD has better connections with Vapnik–Chervonenkis dimension (VCD) [38]. In particular, several new teaching models and complexity measures have been proposed for both the batch teaching settings (e.g., worst-case [18], recursive [22, 24], preference-based [25], and non-clashing models [27]) and the sequential settings (e.g., local preference-based model [26, 29]). These teaching settings, in turn, lead to different notions of TD, that we collectively refer to as LfS-TD. In recent work, [29] has characterized these different notions of TD through a unified framework of modeling learners with preference/ranking functions. In Section 5, we will build on this framework to model the learner's query functions in the LwEQ paradigm through ranking functions, allowing us to connect LwEQ-TD with LfS-TD.

**Teaching in other learning settings.** Within binary classification setting, teaching complexity results have been extended beyond version space learners, including models for gradient learners [39, 40], models inspired by control theory [41, 42], and models for human-centered applications [43, 44, 45, 46]. Furthermore, a recent line of research has studied robust notions of teaching in settings where the teacher has limited information about the learner's dynamics [47, 48, 49]. Given the importance of teacher-learner interactions in many real-world applications, teaching has also been studied in richer domains. In particular, teaching complexity has been investigated for imitation learning settings where the teacher provides demonstrations [50, 51, 52, 53, 54], and for reinforcement learning settings where the teacher provides reward feedback [55, 56]. We see these works as complementary to ours, and we refer the reader to see [28] for an overview.

## 2 Problem Setup

**Teaching framework.** Let $\mathcal{X}$ be a ground set of unlabeled instances and $\mathcal{Y}$ the set of labels. Let $\mathcal{H}$ be a finite class of hypotheses; each element $h \in \mathcal{H}$ is a function $h : \mathcal{X} \to \mathcal{Y}$. Here, we only consider boolean functions, and hence $\mathcal{Y} = \{0, 1\}$. Let $\mathcal{Z} \subseteq \mathcal{X} \times \mathcal{Y}$ be the ground set of labeled examples. Each element $z = (x_z, y_z) \in \mathcal{Z}$ represents a labeled example where the label is given by the target hypothesis $h^*$, i.e., $y_z = h^*(x_z)$. Furthermore, for any $Z \subseteq \mathcal{Z}$, we define *version space* induced by the examples $Z$ as the subset of hypotheses $\mathcal{H}(Z) \subseteq \mathcal{H}$ that are consistent with the labels of all the examples, i.e.,

$$\mathcal{H}(Z) := \{ h \in \mathcal{H} \mid \forall z = (x_z, y_z) \in Z, \ h(x_z) = y_z \} . \tag{1}$$

**Equivalence queries.** We consider the LwEQ paradigm where a learner seeks to identify a target hypothesis from the hypothesis class via equivalence queries. In an equivalence query, the learner asks if the current hypothesis, say $h' \in \mathcal{H}$, is equivalent to the target hypothesis $h^*$ or not. The teacher provides a response $\mathbf{r}$ where $\mathbf{r}$ is either "yes" if $h' \equiv h^*$ or "no" along with a counterexample $z := (x_z, y_z) \in \mathcal{Z}$, such that $h'(x_z) \neq y_z$.

**Learner model and query protocol.** We consider a generic model of the learner that captures our assumptions about how the learner conjectures its hypothesis for equivalence queries based on the responses received from the teacher. A key aspect of this model is the learner's query function $\ell$

over the hypotheses. Based on the information encoded in the inputs of this query function (i.e., the current hypothesis and the history of counterexamples), the learner will choose one hypothesis in $\mathcal{H}$. In the beginning, the learner starts with an initial hypothesis $h_0 \in \mathcal{H}$, the history is $Z_0 = \emptyset$, and the version space is $H_0 = \mathcal{H}$. At a time step $t \geq 1$, the learner first picks the hypothesis $h_t$ as follows:

$$\ell(Z_{t-1}, h_{t-1}) \longrightarrow h_t \in \mathcal{H}(Z_{t-1}), \tag{2}$$

where $Z_{t-1}$ is the history of counterexamples seen up until time $t$ and $\mathcal{H}(Z_{t-1})$ is the corresponding version space. The learner then queries $h_t$ for equivalence to $h^*$ and receives a response $\mathbf{r}_t$. Then, the query protocol proceeds as follows: (i) if $\mathbf{r}_t$ is "yes", the learner has identified $h^*$ and stops; (ii) otherwise $\mathbf{r}_t$ is "no" along with a counterexample $z_t$ using which the learner updates $Z_t = Z_{t-1} \cup \{z_t\}$, and continues. We summarize this query protocol in Algorithm 1.

---

**Algorithm 1:** Query protocol between the learner and the teacher

---
1  Learner's initial hypothesis is $h_0 \in \mathcal{H}$, history is $Z_0 = \emptyset$, and version space is $\mathcal{H}_0 = \mathcal{H}$;
2  **for** $t = 1, 2, 3, \cdots$ **do**
3      learner picks $h_t \in \mathcal{H}(Z_{t-1})$ based on $Z_{t-1}$ and $h_{t-1}$ as per Eq. (2);
4      learner performs an equivalence query with $h_t$;
5      teacher provides a response $r_t$ that is either "yes" or 'no" along with a counterexample $z_t$;
6      **if** $r_t$ *is "yes"* **then**
           |  learner has identified $h^*$ and stops;
7      **else**
           |  learner updates $Z_t = Z_{t-1} \cup \{z_t\}$;

---

We assume that both the learner and the teacher have full knowledge of $\mathcal{X}$, $\mathcal{Y}$, and $\mathcal{H}$; in addition, the teacher knows the target $h^*$ as well as the learner's query function $\ell$. In this work, we consider learner models which could be characterized by a specific query function $\ell$ as discussed above. These include well-known learners studied in the literature, such as a *constant* query learner (denoted as $\ell_{\text{const}}$) who picks the next hypothesis $h_t$ arbitrarily in $\mathcal{H}(Z_{t-1})$ without any preference [18, 1, 29], a *global* query learner (denoted as $\ell_{\text{global}}$) who uses a global ranking over $\mathcal{H}$ to pick the next hypothesis $h_t$ in $\mathcal{H}(Z_{t-1})$ as per Eq. (2) [25, 29], and the Max-Min learning algorithm (denoted as $\ell_{\text{Max-Min}}$) introduced in a recent work on LwEQ paradigm [14].

**Complexity of teaching (i.e., the number of queries made).** In this paper, we study the number of equivalence queries asked by the learner to the teacher to identify a target hypothesis in Algorithm 1, and we call it the *query complexity* or *teaching complexity for LwEQ* paradigm interchangeably. Clearly, this query complexity depends on the learner's query function $\ell$ and the choice of counterexamples by the teacher. In the following sections, we study this complexity for different teacher types depending on the informativeness of the provided counterexamples, as well as for different families of learner types.

## 3 The Query Complexity with Best-Case Teacher: LwEQ-TD

In this section, we consider the optimal teacher who picks best-case counterexamples with the objective of minimizing the learner's queries for identifying $h^*$. For this optimal teacher, we provide a formal characterization of teaching complexity, namely *learning-with-equivalence queries teaching dimension* (LwEQ-TD) paradigm, inspired by different notions of teaching dimension for the learning-from-samples (LfS-TD) paradigm [18, 24, 26, 29].

**Notation.** We denote by $\mathcal{L}$ a family of learner models—alternatively, we can think of them as a family of query functions. To begin, we fix a query function $\ell \in \mathcal{L}$ that the learner uses to pick next hypotheses for equivalence queries. Our characterization below will be based on understanding the minimal "cost" (i.e., the number of queries needed) in steering the learner from a hypothesis $h$ with the history of counterexamples $Z$ to the target hypothesis $h^*$—in Algorithm 1, $h$ refers to $h_{t-1}$ and $Z$ refers to $Z_{t-1}$ at the beginning of time $t$.

**Minimal cost of steering.** We begin by providing a recursive function that captures this cost of steering and will be key to formalize teaching complexity in the LwEQ paradigm. As is typically considered in the LfS paradigm [18, 24, 26, 29], we consider the "adversarial" perspective in how the learner

breaks ties in picking the hypothesis w.r.t. its query function in Eq. (2). The optimal cost for steering the learner from the current history $(Z, h)$ to $h^*$ in the query protocol of Algorithm 1 is then given by:

$$D_\ell(Z, h, h^*) = \begin{cases} 0, & \text{if } \ell(Z, h) = \{h^*\} \\ 1 + \max_{h' \in \ell(Z,h)} \min_{z:h'(x_z) \neq y_z} D_\ell(Z \cup \{z\}, h', h^*), & \text{otherwise} \end{cases} \tag{3}$$

Note that $D_\ell$ in Eq. (3) has a $\max$ operator w.r.t. the learner's choice and $\min$ operator with the teacher's choice (see [29]). Furthermore, in this function, we are not counting the last query when the learner's queried hypothesis $h_t$ equals $h^*$ (hence, the number of queries is same as the number of counterexamples received by the learner).

**LwEQ-TD.** Given a fixed query function $\ell \in \mathcal{L}$, an initial hypothesis $h_0$, and a target hypothesis $h^*$, LwEQ-TD w.r.t. $\ell$ and $h^*$ is the optimal cost for teaching the target hypothesis $h^*$:

$$\text{LwEQ-TD}_{\mathcal{X}, \mathcal{H}, h_0}(\ell, h^*) = D_\ell(\emptyset, h_0, h^*). \tag{4}$$

To characterize the teaching complexity for the hypothesis class, we consider the worst-case target hypothesis in $\mathcal{H}$, given by:

$$\text{LwEQ-TD}_{\mathcal{X}, \mathcal{H}, h_0}(\ell) = \max_{h^*} D_\ell(\emptyset, h_0, h^*). \tag{5}$$

Finally, to compare LwEQ-TD with existing notions of TD in the LfS paradigm (i.e., LfS-TD, see Footnote 1), we define LwEQ-TD for a given family of query functions $\mathcal{L}$. Based on [24, 27, 29], we define LwEQ-TD w.r.t the family $\mathcal{L}$ as the teaching complexity w.r.t. the best $\ell$ in that family:

$$\text{LwEQ-TD}_{\mathcal{X}, \mathcal{H}, h_0}(\mathcal{L}) = \min_{\ell \in \mathcal{L}} \max_{h^*} D_\ell(\emptyset, h_0, h^*). \tag{6}$$

In the following two sections, we will investigate how LwEQ-TD (the complexity of the optimal teacher in the LwEQ paradigm) compares with various other complexity notions. In Section 4, we investigate different teachers in the LwEQ paradigm, showcasing the power of the optimal teacher. In Section 5, we establish new connections of LwEQ-TD with LfS-TD.

## 4  The Query Complexity for Different Teachers in the LwEQ Paradigm

In this section, we study the query complexity in the LwEQ paradigm for different types of teachers. In Section 3, we characterized the query complexity for the optimal teacher who provides best-case counterexamples. The goal of this section is to compare the query complexity for the optimal teacher with other variants of teachers as discussed below. We will denote the teacher as TEQ, and we consider the following four variants of teachers:

- best-TEQ responds "yes" or "no" along with best-case counterexamples (see Section 3).
- random-TEQ responds "yes" or "no" along with random counterexamples (picked uniformly at random). As discussed in [14], we characterize the query complexity for random-TEQ as the expected number of counterexamples provided by random-TEQ in Algorithm 1.
- worst-TEQ responds "yes" or "no" along with worst-case (least informative) counterexamples. We characterize query complexity for worst-TEQ by replacing the $\min$ over the choice of counterexamples with a $\max$ in the function $D_\ell$ in Eq. (3).
- binary-TEQ responds "yes" or "no" without any counterexamples.

In the rest of the section, we compare the query complexity for these teachers when interacting with different learners and we characterize this complexity by the richness of their query functions.

### 4.1  Warm-up: Query Complexity Bounds When Teaching Different Types of Learners

In this section, we study the query complexity bounds for different teachers when teaching various learner models characterized by the richness of the underlying query functions. In particular, we consider the query functions $\ell_{\text{const}}$, $\ell_{\text{global}}$, and $\ell_{\text{Max-Min}}$ (see Section 2). $\ell_{\text{const}}$ picks the next hypothesis $h_t$ arbitrarily in $\mathcal{H}(Z_{t-1})$ without any preference [29]. $\ell_{\text{global}}$ uses a global ranking over

$\mathcal{H}$ to pick the next hypothesis $h_t$ in $\mathcal{H}(Z_{t-1})$ as per Eq. (2) and is popularly studied in the LfS paradigm [25, 29]. $\ell_{\text{Max-Min}}$ uses a richer history-dependent query function, and was introduced for achieving a better query complexity when the counterexamples are chosen randomly [14]. To study the query complexity for different settings (corresponding to four teachers and three learners), we consider the hypothesis class of Axes-aligned hyperplanes, a simple yet rich hypothesis class, that generalizes the canonical hypothesis class of threshold boolean functions to higher dimensions. Next, we introduce this hypothesis class.[2]

**Definition 4.1** (Axes-aligned hyperplanes). *Fix an input space $\mathcal{X} = \{1, 2, \ldots, n\}^d$ in $\mathbb{R}^d$ with labels set $\mathcal{Y} = \{0, 1\}$, where an input $\mathbf{x} \in \mathcal{X}$ is a $d$-dimensional point in $\mathbb{R}^d$, i.e., $\mathbf{x} := (x_1, x_2, \ldots, x_d)$ such that all $x_i \in [1, n+1]$. We define the hypothesis class of Axes-aligned hyperplanes as:*

$$\mathcal{H}_{\text{axh}} := \{h \mid \exists\, i \in [1, d+1],\ j \in [n+1],\ \text{s.t.}\ \forall \mathbf{x} \in \mathcal{X},\ h(\mathbf{x}) = 1 \text{ if } x_i \leq j,\ \text{otherwise } 0\}. \quad (7)$$

We summarize the query complexity bounds for $\ell_{\text{const}}$, $\ell_{\text{global}}$, and $\ell_{\text{Max-Min}}$ under different teaching scenarios in Table 2. We note that $|\mathcal{H}_{\text{axh}}| = d \cdot (n+1)$. To capture the global ranking of $\ell_{\text{global}}$, we use a ranking function $g : \mathcal{H}_{\text{axh}} \to \{0, \ldots, |\mathcal{H}_{\text{axh}}| - 1\}$. For a target hypothesis $h^* \in \mathcal{H}_{\text{axh}}$ and a ranking function $g$, the bounds for $\ell_{\text{global}}$ in Table 2 are described using the following two quantities: (i) $t_g^* := \left|\{h \in \mathcal{H}_{\text{axh}} \text{ s.t. } g(h) \leq g(h^*)\}\right|$ and (ii) $n_g^*$ is defined as the highest number of hypotheses aligned along one of the axis that are preferred over $h^*$ as per the ranking function $g$. Note that $1 \leq t_g^* \leq d \cdot (n+1)$ and $1 \leq n_g^* \leq n+1$. In the Appendix of the supplementary, we state the bounds from Table 2 as theorems and provide detailed proofs.

| Learner \\ Teacher | binary-TEQ (Yes/No) | worst-TEQ | random-TEQ | best-TEQ |
|---|---|---|---|---|
| $\ell_{\text{const}}$ | $d \cdot (n+1)$ | $\Omega\left(d \cdot (n+1)\right)$ | $\mathcal{O}\left(d \cdot \log(n+1)\right)$ | 2 |
| $\ell_{\text{global}}$ | $t_g^*$ | $\Omega\left(t_g^*\right)$ | $\mathcal{O}\left(d \cdot \log n_g^*\right)$ | 1 |
| $\ell_{\text{Max-Min}}$ | (Not-Applicable) | $\Omega\left(d \cdot \log(n+1)\right)$ | $\mathcal{O}\left(\log(d \cdot (n+1))\right)$ | 2 |

Table 2: Query complexity for the Axes-aligned hyperplanes considering different teachers and learners; see Section 4.1 for details.

## 4.2 Lower and Upper Bounds on Query Complexity

To establish lower bounds on the query complexity for three teachers (binary-TEQ, worst-TEQ, random-TEQ), we will consider learner models characterized with *optimal query functions*. In contrast to these lower bounds, we establish upper bounds on the query complexity for the best-case teacher (best-TEQ) by considering a weaker learner model characterized with a global query function (see Section 2). To study the bounds in these different teaching settings, we consider two more hypothesis classes beyond Axes-aligned hyperplanes: (i) Monotone monomials [18, 37] and (ii) Orthogonal rectangles [31, 18, 57, 58], that have been extensively studied in the LfS paradigm.

We summarize the results in Table 3 with detailed proofs deferred to the Appendix of the supplementary. First, we state the results on query complexity for Axes-aligned hyperplanes in the theorem below.

**Theorem 1.** *Consider the hypothesis class of Axes-aligned hyperplanes $\mathcal{H}_{\text{axh}}$ (see Eq. (7)). There exists a global learner $\ell_{\text{global}}$ such that best-TEQ achieves LwEQ-TD($\ell_{\text{global}}$) of exactly 1. In contrast, for any optimal learner, random-TEQ provides at least $\Omega(\log d)$ counterexamples "in expectation" and worst-TEQ provides at least $\Omega(d + \log(n+1))$ counterexamples "in the worst-case".*

The results in the above theorem are based on the following key ideas. The bound for best-TEQ is based on results from Section 4.1, where we showed that a specific choice of $\ell_{\text{global}}$ upper bounds the query complexity for best-TEQ by 1. On the other hand, as per Theorem 25 (see [14]), the query complexity for a random teacher is connected to the VC dimension of a hypothesis class independent of the learner model. We show that $\text{VCD}(\mathcal{H}_{\text{axh}})$ is $\Omega(\log d)$, which entails a direct lower bound on the query complexity for random-TEQ for any learner model. Furthermore, we note that the

---

[2]Given two positive integers $a$ and $b$ where $a < b$, we use the following shorthand notation: $[a, b] = \{a, a+1, \ldots, b-1\}$ and $[b] = \{0, 1, \ldots, b-1\}$.

| Hypothesis class \ Teacher | binary-TEQ (Yes/No) | worst-TEQ | random-TEQ | best-TEQ |
|---|---|---|---|---|
| Axes-aligned hyperplanes | $\Omega\left(d \cdot (n+1)\right)$ | $\Omega\left(d + \log(n+1)\right)$ | $\Omega\left(\log d\right)$ | 1 |
| Monotone monomials | $\Omega\left(2^n\right)$ | $\Omega\left(n\right)$ | $\Omega\left(n\right)$ | 1 |
| Orthogonal rectangles | $\Omega\left((n \cdot (n+1))^d\right)$ | $\Omega\left(d \cdot \log(n+1)\right)$ | $\Omega\left(d\right)$ | 2 |

Table 3: Lower and upper bounds on query complexity for the Axes-aligned hyperplanes, Monotone monomials, and Orthogonal rectangles when considering different teaching settings. The lower bounds for three teachers (binary-TEQ, worst-TEQ, random-TEQ) are established based on learner models characterized with *optimal query functions*; the upper bounds for the best-case teacher (best-TEQ) are established based on learner models characterized with a global query function. Additional details are provided in the proofs of Theorems 1, 2, and 3.

worst-TEQ could force any learner to query a hypothesis in every axis in addition to $\Omega\left(\log(n+1)\right)$ queries in the axis of the target hypothesis.

Now, we define the hypothesis class of Monotone monomials and then state the aforementioned query complexity bounds for this class in Theorem 2.

**Definition 4.2** (Monotone monomials). *Fix a set of literals $\{v_1, v_2, \ldots, v_n\}$, an input space $\mathcal{X} = \{0,1\}^n$, and labels set $\mathcal{Y} = \{0,1\}$. A monotone monomial is a negation-free conjunction of literals. If $mono(i_1, i_2, \ldots, i_k)$ denotes a monotone monomial over the set of $k$ literals $\{v_{i_1}, v_{i_2}, \ldots, v_{i_k}\}$, then it canonically represents a hypothesis $h(\mathbf{x}) := x_{i_1} \wedge x_{i_2} \wedge \ldots \wedge x_{i_k}$ over the input space. With these notations, we define the hypothesis class of Monotone monomials as:*

$$\mathcal{H}_{\mathrm{mono}} := \left\{ h \,\middle|\, \exists\ \{i_1, i_2, \ldots, i_k\} \subseteq \{1, 2, \ldots, n\},\ \text{s.t.}\ \forall \mathbf{x} \in \mathcal{X},\ \ h(\mathbf{x}) = x_{i_1} \wedge x_{i_2} \wedge \ldots \wedge x_{i_k} \right\}. \tag{8}$$

**Theorem 2.** *Consider the hypothesis class of Monotone monomials $\mathcal{H}_{\mathrm{mono}}$ (see Eq. (8)). There exists a global learner $\ell_{\mathrm{global}}$ such that best-TEQ achieves LwEQ-TD($\ell_{\mathrm{global}}$) of exactly 1. In contrast, for any optimal learner, random-TEQ provides at least $\Omega\left(n\right)$ counterexamples "in expectation" and worst-TEQ provides at least $\Omega\left(n\right)$ counterexamples "in the worst-case".*

Now, we define the hypothesis class of Orthogonal rectangles and then state the aforementioned query complexity bounds for this class in Theorem 3.

**Definition 4.3** (Orthogonal rectangles). *Fix an input space $\mathcal{X} = \{1, \ldots, n\}^d$ in $\mathbb{R}^d$ with labels set $\mathcal{Y} = \{0,1\}$, where an input $\mathbf{x} \in \mathcal{X}$ is a $d$-dimensional point in $\mathbb{R}^d$, i.e, $\mathbf{x} := (x_1, x_2, \ldots, x_d)$ such that all $x_i \in [1, n+1]$. We define the class of Orthogonal rectangles as:*

$$\mathcal{H}_{\mathrm{rec}} := \left\{ h \,\middle|\, \exists\ \{a_j, b_j\}_{j \in [1, d+1]} \subset [n+1],\ \text{s.t.}\ \forall \mathbf{x} \in \mathcal{X},\ \ h(\mathbf{x}) = \begin{cases} 1 & \text{if}\ \ \forall j,\ a_j < x_j \le b_j \\ 0 & \text{otherwise.} \end{cases} \right\}. \tag{9}$$

**Theorem 3.** *Consider the hypothesis class of Orthogonal rectangles $\mathcal{H}_{\mathrm{rec}}$ (see Eq. (9)). There exists a global learner $\ell_{\mathrm{global}}$ such that best-TEQ achieves LwEQ-TD($\ell_{\mathrm{global}}$) of exactly 2. In contrast, for any optimal learner, random-TEQ provides at least $\Omega\left(d\right)$ counterexamples "in expectation" and worst-TEQ provides at least $\Omega\left(d \cdot \log(n+1)\right)$ counterexamples "in the worst-case".*

Similar to Axes-aligned hyperplanes (see Theorem 1), the lower bound results are significantly worse than the upper bound results for rich classes of Monotone monomials (see Theorem 2) and Orthogonal rectangles (see Theorem 3), thereby establishing the power of the best-case teacher. These results are summarized in Table 3.

## 5 Teaching Dimensions for the LwEQ and LfS Paradigms

In this section, we draw new comparisons of the notion of LwEQ-TD in the LwEQ paradigm to existing notions of TD in the LfS paradigm. Based on the teaching setting and the learner models, one gets different notions of teaching complexity and we collectively refer to these notions as LfS-TD (see Footnote 1) [18, 21, 22, 25, 27, 59, 29]. For comparisons with LfS-TD below, we consider the

framework of [29] that provides a unified view of teaching settings by modeling the learners through preference/ranking functions. In the following, we first introduce a notation for these ranking functions $\sigma$ and then discuss how a ranking function $\sigma$, in turn, induces a learner in the LwEQ paradigm.

**Learner's query function $\ell$ using a framework of a ranking $\sigma$.** Consider a hypothesis class $\mathcal{H}$, a version space $H \subseteq \mathcal{H}$, and hypotheses $h', h'', h \in \mathcal{H}$ such that $h', h'' \in H$. Building on the notation for a preference function $\sigma$ (see [29]), we define a ranking function $\sigma : \mathcal{H} \times 2^{\mathcal{H}} \times \mathcal{H} \to \mathbb{R}$ where $\sigma(h'; H, h)$ signifies how $h'$ is ranked in the version space $H$ from the current hypothesis $h$. Thus, we say $h'$ is *ranked* (or preferred) over $h''$ in the current version space $H$ from the current hypothesis $h$ if $\sigma(h'; H, h) \leq \sigma(h''; H, h)$, and vice versa. Similar to preference-based learners [29], a learner's query function (see Section 2) could use this ranking to pick the most preferred hypothesis for an equivalence query. In this section, we consider query functions for our learner model $\ell$ (see Section 2) which use a framework of a ranking function $\sigma$ to pick the next hypothesis $h_t$ based on the current history of counterexamples seen $Z_{t-1}$ and the current hypothesis $h_{t-1}$ as follows:

$$\ell(Z_{t-1}, h_{t-1}) \longrightarrow h_t \in \underset{h' \in \mathcal{H}(Z_{t-1})}{\arg\min} \ \sigma(h'; \mathcal{H}(Z_{t-1}), h_{t-1}). \tag{10}$$

In this section, we identify a learner model $\ell_{\sigma}$[3] with the corresponding ranking $\sigma$ and consequently use the notation LwEQ-TD$_{\mathcal{X}, \mathcal{H}, h_0}(\sigma)$ (for fixed $\mathcal{X}, \mathcal{H}, h_0$) for LwEQ-TD (see Eq. (5), Section 3) for the learner model $\ell_{\sigma}$. We denote a family of ranking functions $\sigma$ as $\Sigma$. For a family of ranking functions $\Sigma$, the corresponding LwEQ teaching dimension is denoted as LwEQ-TD$_{\mathcal{X}, \mathcal{H}, h_0}(\Sigma)$ (see Eq. (6), Section 3).

For families of learner models induced by specific types of ranking functions, we connect LwEQ-TD to existing notions of LfS-TD. In the following, we consider ranking functions broadly categorized into two classes: (i) ranking functions independent of $Z_{t-1}$ and $h_{t-1}$; (ii) ranking functions dependent on $Z_{t-1}$ and/or $h_{t-1}$.

## 5.1 LwEQ Learners with Ranking Functions Independent of $Z_{t-1}$ and $h_{t-1}$

These ranking functions induce learners whose next equivalence query at time $t$ (i.e., choice of the hypothesis $h_t$) is independent of the history of counterexamples $Z_{t-1}$ and independent of the hypothesis $h_{t-1}$ (see Algorithm 1, Section 2). In Section 4.1, we discussed these learner families under the name of *contant* and *global* learners, and we formalize these families below using the ranking functions framework of Eq. (10). In particular, we introduce two families of ranking functions: (i) $\Sigma_{\text{const}}$ is a family of *constant* ranking functions where $\sigma \in \Sigma_{\text{const}}$ ranks every hypothesis equally without any preference; (ii) $\Sigma_{\text{global}}$ is a family of *global* ranking functions where $\sigma \in \Sigma_{\text{global}}$ ranks hypothesis based on a global preference. These two families are given below:

$$\Sigma_{\text{const}} = \{\sigma \mid \exists \, c \in \mathbb{R}, \text{ s.t. } \forall h', Z, h, \ \sigma(h'; \mathcal{H}(Z), h) = c\}. \tag{11}$$

$$\Sigma_{\text{global}} = \{\sigma \mid \exists \, g : \mathcal{H} \to \mathbb{R}, \text{ s.t. } \forall h', Z, h, \ \sigma(h'; \mathcal{H}(Z), h) = g(h')\}. \tag{12}$$

Given these ranking functions, the following theorem establishes the connection of LwEQ-TD$_{\mathcal{X}, \mathcal{H}, h_0}(\Sigma_{\text{const}})$ and LwEQ-TD$_{\mathcal{X}, \mathcal{H}, h_0}(\Sigma_{\text{global}})$ with existing notions of LfS-TD; also see Eq. (6) in Section 3.

**Theorem 4.** *Fix $\mathcal{X}, \mathcal{H}, h_0$. For learners whose query function is induced by a ranking function independent of $Z_{t-1}$ and $h_{t-1}$, the corresponding LwEQ-TD is equivalent to the notions of LfS-TD as follows:*

$$\text{LwEQ-TD}_{\mathcal{X}, \mathcal{H}, h_0}(\Sigma_{\text{const}}) = \text{wc-TD}(\mathcal{H}). \tag{13}$$

$$\text{LwEQ-TD}_{\mathcal{X}, \mathcal{H}, h_0}(\Sigma_{\text{global}}) = \text{RTD}(\mathcal{H}). \tag{14}$$

When the rankings are independent of $Z_{t-1}$ and $h_{t-1}$, Eq. (13) and Eq. (14) connect LwEQ-TD to the notions of wc-TD [18] and RTD [22, 24] in batch teaching settings (the optimal teaching sequence is invariant to its permutation) of the LfS paradigm. To show the equalities, we note that if $Z' \subseteq \mathcal{Z}$ is a teaching sequence for fixed $h^*$ in the LfS paradigm, then there exists a permutation of $Z'$ which forms a teaching sequence of counterexamples for the learner in the LwEQ

---

[3]We use the notation $\ell_{\sigma}$ for a learner's query function $\ell$ using a ranking $\sigma$.

paradigm. Furthermore, this theorem allows us to connect LwEQ-TD$_{\mathcal{X},\mathcal{H},h_0}(\Sigma_{\text{global}})$ with VCD$(\mathcal{H})$ as RTD$(\mathcal{H}) = \mathcal{O}\left(\text{VCD}^2(\mathcal{H})\right)$ [24, 29]. A detailed proof of the theorem is in the Appendix of the supplementary.

## 5.2 LwEQ Learners with Ranking Functions dependent on $Z_{t-1}$ and/or $h_{t-1}$

Here, we consider ranking functions that induce learners whose next equivalence query at time $t$ (i.e., choice of the hypothesis $h_t$) is dependent on the history of counterexamples $Z_{t-1}$ and/or on the hypothesis $h_{t-1}$ (see Algorithm 1, Section 2). We formalize these families below using the ranking functions framework of Eq. (10). In particular, we introduce three families of ranking functions: (i) $\Sigma_{\text{gvs}}$ is a family of *global version space* ranking functions where $\sigma \in \Sigma_{\text{gvs}}$ ranks hypotheses based on a global preference dependent on $Z_{t-1}$ but independent of $h_{t-1}$; (ii) $\Sigma_{\text{local}}$ is a family of *local* ranking functions where $\sigma \in \Sigma_{\text{local}}$ ranks hypotheses based on a local preference dependent only on $h_{t-1}$; (iii) $\Sigma_{\text{lvs}}$ is a family of *local version space* ranking functions where $\sigma \in \Sigma_{\text{lvs}}$ ranks hypotheses based on a local preference dependent on $Z_{t-1}$ and $h_{t-1}$. These three families are given below:

$$\Sigma_{\text{gvs}} = \left\{ \sigma \,\middle|\, \exists\, g : \mathcal{H} \times 2^{\mathcal{H}} \to \mathbb{R}, \text{ s.t. } \forall h', Z, h, \ \sigma(h'; \mathcal{H}(Z), h) = g(h', \mathcal{H}(Z)) \right\}. \tag{15}$$

$$\Sigma_{\text{local}} = \left\{ \sigma \,\middle|\, \exists\, g : \mathcal{H} \times \mathcal{H} \to \mathbb{R}, \text{ s.t. } \forall h', Z, h, \ \sigma(h'; \mathcal{H}(Z), h) = g(h', h) \right\}. \tag{16}$$

$$\Sigma_{\text{lvs}} = \left\{ \sigma \,\middle|\, \exists\, g : \mathcal{H} \times 2^{\mathcal{H}} \times \mathcal{H} \to \mathbb{R}, \text{ s.t. } \forall h', Z, h, \ \sigma(h'; \mathcal{H}(Z), h) = g(h', \mathcal{H}(Z), h) \right\}. \tag{17}$$

As discussed in the LfS paradigm, these ranking functions could lead to the learner and the teacher colluding to achieve arbitrarily low teaching complexity [29]. To avoid this, we consider a specific collusion-free behavior where the ranking is consistent with its choice of hypothesis. More formally,

**Definition 5.1 (Collusion-free ranking [29]).** *Consider a time $t$ where the learner's current hypothesis is $h_{t-1}$ and the history of inputs seen is $Z_{t-1}$. Further assume that the learner's preferred hypothesis for time $t$ is uniquely given by* $\arg\min_{h' \in \mathcal{H}(Z_{t-1})} \sigma(h'; \mathcal{H}(Z_{t-1}), h_{t-1}) = \{\hat{h}\}$*. Let $S$ be additional examples provided by an adversary from time $t$ onwards. We call a ranking function $\sigma$ collusion-free if for any $S$ consistent with $\hat{h}$, it holds that* $\arg\min_{h' \in \mathcal{H}(Z_{t-1} \cup S)} \sigma(h'; \mathcal{H}(Z_{t-1} \cup S), \hat{h}) = \{\hat{h}\}$*.*

For the ranking functions in Eqs. (15)-(17), we consider subsets that satisfy Definition 5.1, and consequently the corresponding collusion-free families are denoted as $\Sigma_{\text{gvs}}^{\text{CF}}$, $\Sigma_{\text{local}}^{\text{CF}}$, and $\Sigma_{\text{lvs}}^{\text{CF}}$. For these ranking functions, we study the connection of LwEQ-TD$_{\mathcal{X},\mathcal{H},h_0}(\Sigma_{\text{gvs}}^{\text{CF}})$, LwEQ-TD$_{\mathcal{X},\mathcal{H},h_0}(\Sigma_{\text{local}}^{\text{CF}})$, and LwEQ-TD$_{\mathcal{X},\mathcal{H},h_0}(\Sigma_{\text{lvs}}^{\text{CF}})$ with existing notions of LfS-TD, in particular NCTD [27], local-PBTD [59, 29], and wc-TD [18, 29]. We state these connections in the following theorem and provide detailed proofs in the Appendix of the supplementary.

**Theorem 5.** *Fix $\mathcal{X}, \mathcal{H}, h_0$. For learners whose query function is induced by a ranking function dependent on $Z_{t-1}$ and/or $h_{t-1}$, the corresponding LwEQ-TD is connected to the notions of LfS-TD as:*

$$\text{LwEQ-TD}_{\mathcal{X},\mathcal{H},h_0}\left(\Sigma_{\text{gvs}}^{\text{CF}}\right) = \text{NCTD}(\mathcal{H}). \tag{18}$$

$$\text{local-PBTD}_{\mathcal{X},\mathcal{H},h_0} \leq \text{LwEQ-TD}_{\mathcal{X},\mathcal{H},h_0}(\Sigma_{\text{local}}^{\text{CF}}) \leq \text{wc-TD}(\mathcal{H}). \tag{19}$$

$$\text{lvs-PBTD}_{\mathcal{X},\mathcal{H},h_0} \leq \text{LwEQ-TD}_{\mathcal{X},\mathcal{H},h_0}(\Sigma_{\text{lvs}}^{\text{CF}}) \leq \text{wc-TD}(\mathcal{H}). \tag{20}$$

To achieve equality in Eq. (18), we observe that the family of rankings $\Sigma_{\text{gvs}}^{\text{CF}}$ leads to a batch teaching setting in the LfS paradigm. To show this equality, we establish the permutation invariance of a teaching set $Z \subseteq \mathcal{Z}$ in the LfS paradigm to form a teaching sequence of counterexamples in the LwEQ paradigm using the collusion-freeness property of the underlying ranking function.

To show the lower bounds in Eq. (19) and Eq. (20), we note that a teaching sequence of counterexamples in the LwEQ paradigm forms a teaching sequence in the LfS paradigm. For the upper bound of wc-TD$(\mathcal{H})$, we note that a *constant* query learner (see Eq. (11)) could pick any consistent hypothesis in a version space to maximize the number of counterexamples. This observation, along with the results in Theorem 4, leads to the desired upper bound.

# 6    Concluding Discussions

We investigated the query complexity for the learning-with-equivalence-queries (LwEQ) paradigm when the counterexamples are provided by the optimal teacher. We introduced LwEQ-TD, a notion of teaching dimension, to characterize the complexity of teaching (i.e., the number of queries made) for the optimal teacher. We showed the power of best-case counterexamples picked by the optimal teacher, in contrast to worst-case or random counterexamples, for different hypothesis classes, including Axes-aligned hyperplanes, Monotone monomials, and Orthogonal rectangles. We further established new connections of LwEQ-TD with existing notions of TD in the learning-with-samples paradigm, including wc-TD, RTD, NCTD, and local-PBTD.

In the learning-with-queries paradigm, several works have analyzed the query complexity for a combination of different query types [1, 8, 15], such as membership queries, equivalence queries, among others. Building on our characterization of LwEQ-TD, an important research direction of future work is to investigate similar notions of TD when a learner can ask a combination of different queries. Alternatively, one could study LwEQ in the setting of *improper* equivalence queries where the learner can pick the queried hypothesis outside the hypothesis class. [31] considered improper equivalence queries in the LwEQ paradigm, leading to a reduction in query complexity for various hypothesis classes. It would be important to characterize the teaching complexity for best-case counterexamples with improper equivalence queries.

Another important research direction is to further investigate LwEQ-TD for more complex hypothesis classes, including DFA (deterministic finite automaton), NFA (nondeterministic finite-state acceptors), CFG (context-free grammars), among others. As a concrete hypothesis class, one could consider $DFA_s^2$ (i.e., DFA with alphabet size 2 and state size $s$). For this hypothesis class, the query complexity is exponential in $s$ when considering worst-case counterexamples [13]. [60] computed VCD for a variety of hypothesis classes including $DFA_s^2$; this result along with query complexity bounds of the Max-Min algorithm in [14] establishes the bound of $\Theta\left(s \log s\right)$ on the expected number of random counterexamples for $DFA_s^2$. As future work, it would be interesting to investigate LwEQ-TD for $DFA_s^2$. A concrete direction is to study the query complexity when providing more structured counterexamples (e.g., by picking minimal length counterexamples as considered in [12]) as this would allow establishing an upper bound for best-case counterexamples.

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
