# A    List of Appendices

Now, we list the appendices which provide the proofs of our theoretical results in full detail. The appendices are summarized as follows:

- Appendix B discusses different types of learners and contains proofs of the formal results in Section 4.1.
  - Appendix B.1 discusses learner models characterized by query functions $\ell_{\text{const}}$, $\ell_{\text{global}}$, and $\ell_{\text{Max-Min}}$.
  - Appendix B.2 defines the hypothesis class of Threshold functions and provides query complexity bounds for different teachers when teaching learner models characterized by query functions $\ell_{\text{const}}$, $\ell_{\text{global}}$, and $\ell_{\text{Max-Min}}$.
  - Appendix B.3 contains proofs of the formal results shown in Table 2 for Axes-aligned hyperplanes.
- Appendix C contains proofs of the formal results as shown in Table 3 in Section 4.2.
  - The proof of Theorem 1 is in Appendix C.1.
  - The proof of Theorem 2 is in Appendix C.2.
  - The proof of Theorem 3 is in Appendix C.3.
- Appendix D contains proofs of the formal results in Section 5.
  - The proof of Theorem 4 is in Appendix D.1.
  - The proof of Theorem 5 is in Appendix D.2.

# B  Query Complexity Bounds When Teaching Different Types of Learners (Section 4.1)

In this appendix, we provide the proofs for the bounds shown in Table 2 on the query complexity for Axes-aligned hyperplanes (Definition 4.1). We divide the appendix as follows: Appendix B.1 discusses learner models characterized by query functions $\ell_{\text{const}}$, $\ell_{\text{global}}$, and $\ell_{\text{Max-Min}}$; Appendix B.2 discusses Threshold functions and their query complexity bounds for different teachers; Appendix B.3 provides the proofs to entries in Table 2.

## B.1  Learner Types

In the following, we discuss learners characterized by the query functions $\ell_{\text{const}}$, $\ell_{\text{global}}$, and $\ell_{\text{Max-Min}}$ used for the query complexity bounds in Table 2.

**Constant query learner ($\ell_{\text{const}}$).**  A constant query learner ($\ell_{\text{const}}$) picks the next hypothesis $h_t$ arbitrarily in $\mathcal{H}(Z_{t-1})$ without any preference [18, 1, 29]. To analyze the query complexity for different hypothesis classes we assume that the learner picks the worst-case hypothesis in $\mathcal{H}(Z_{t-1})$ to query for equivalence. Since all the constant query learners have the same query function (see Section 2) the query complexity bounds are identified for the entire family of constant learners. We characterize a family of constant query learners in Section 5.

**Global query learner ($\ell_{\text{global}}$).**  A global query learner ($\ell_{\text{global}}$) uses a global ranking over $\mathcal{H}$ to pick the next hypothesis $h_t$ in $\mathcal{H}(Z_{t-1})$ as per Eq. (2) [25, 29]. We denote the global ranking of $\ell_{\text{global}}$ using a function $g : \mathcal{H} \to [|\mathcal{H}|]$. Similar to a constant learner $\ell_{\text{const}}$, we assume that $\ell_{\text{global}}$ picks the worst ranked hypothesis according to the function $g$ for querying (ties are broken arbitrarily). We characterize a family of global query learners in Section 5.

**Max-Min query learner ($\ell_{\text{Max-Min}}$).**  In the LwEQ paradigm, [14] considered the setting where a learner queries a hypothesis for equivalence and receives a random counterexample picked from a known probability distribution and introduced the Max-Min learning algorithm. A Max-Min query learner picks the next hypothesis $h_t$ based on the current history of counterexamples $Z_{t-1}$ but *agnostic* of the current hypothesis $h_{t-1}$ as follows:

$$h_t \in \underset{h \in \mathcal{H}(Z_{t-1})}{\arg\max} \min_{h' \in \mathcal{H}(Z_{t-1}) \setminus \{h\}} E(h, h') \tag{21}$$

where $E(h, h')$ is given by the expected fraction[4] of hypothesis eliminated from $\mathcal{H}(Z_{t-1})$ if an equivalence query is posed with hypothesis $h$ and target hypothesis $h'$, assuming counterexamples are picked from a known probability distribution. In this work, we assume that the underlying probability distribution is uniform. Furthermore, to evaluate a Max-Min query learner in the worst-case teaching scenario (similar for best-case teaching), i.e when worst-TEQ provides counterexamples (see *Column 2* in Table 2) we assume that the learner computes $E(h, h')$ modeling a uniform distribution over the set of valid counterexamples. We discuss the query complexity bounds for $\ell_{\text{Max-Min}}$ under different teaching scenarios in Appendix B.3.

## B.2  Threshold Functions

Now, we discuss Threshold functions and study their query complexity bounds under different teaching scenarios in the LwEQ paradigm. Results and insights gained in this section would be used in many other results in our work. First, we define the hypothesis class of Threshold functions as follows:

**Definition B.1** (Threshold functions). *Fix an input space $\mathcal{X} = \{1, \cdots, n\}$ in $\mathbb{R}$ with labels set $\mathcal{Y} = \{0, 1\}$. We define the hypothesis class of Threshold functions as:*

$$\mathcal{H}_{\text{threshold}} := \{h \mid \exists j \in [n+1], \text{ s.t. } \forall x \in \mathcal{X}, \ h(x) = 1 \text{ if } x \leq j, \text{ otherwise } 0\}. \tag{22}$$

---

[4]Query function remains the same if we compute the expected number, i.e. expected fraction $\times$ total number of hypotheses in the version space.

We note that if a teacher provides two counterexamples corresponding to the threshold $j$ of a target hypothesis $h^* \in \mathcal{H}_{\text{threshold}}$, then it eliminates any other hypothesis in the class. Thus, it is straightforward that LwEQ-TD$_{\mathcal{X}, \mathcal{H}_{\text{threshold}}, h_0}(\ell_{\text{const}}) = 2$. On the other hand, providing just the rightmost example classified as 1 is sufficient for a global learner $\ell_{\text{global}}$, i.e. LwEQ-TD$_{\mathcal{X}, \mathcal{H}_{\text{threshold}}, h_0}(\ell_{\text{global}}) = 1$.

In the following theorem, we establish query complexity bounds for worst-TEQ and random-TEQ when teaching a constant query learner $\ell_{\text{const}}$.

**Theorem 6.** *Consider the hypothesis space of $\mathcal{H}_{\text{threshold}}$ over the input space $\mathcal{X}$. Fix a constant query learner $\ell_{\text{const}}$. "In the worst-case", $\ell_{\text{const}}$ queries at least[5] $\Omega(n+1)$ times to worst-TEQ. Furthermore, $\ell_{\text{const}}$ queries at most $\mathcal{O}(\log(n+1))$ times "in expectation" to random-TEQ.*

*Proof.* In the worst-case teaching scenario, it is easy to note that worst-TEQ acts as an adversary and provides counterexamples chosen from the leftmost examples. Thus, "in the worst-case" $\ell_{\text{const}}$ queries at least $\Omega(n+1)$ times to identify the target.

Now, we show the bound for random-TEQ when teaching a constant learner $\ell_{\text{const}}$. For random-TEQ which provides random counterexamples (sampled uniformly) to the queried hypothesis, we first show that the expected number of counterexamples has the form $\sum_{i=1}^{n} \frac{1}{i}$, which in turn establishes the bound. Consider the random variable $X_n$ to be the number of counterexamples given by random-TEQ for the successful identification of a target hypothesis. Define $W(n)$ to be the expected number of counterexamples provided "in the worst-case" to $\ell_{\text{const}}$ for Threshold functions $\mathcal{H}_{\text{threshold}}$. Thus, we note $W(n) = \mathbb{E}[X_n]$. Using induction, we show that $W(n) = \sum_{i=1}^{n} \frac{1}{i}$. Note, $W(1) = 1$ and $W(2) = 1 + \frac{1}{2}$. Assume the induction statement for $n = k$. Now, we would prove the statement for $n = k + 1$. Consider the following:

$$W(k+1)$$

$$= \sum_{i=1}^{k+1} i \cdot \mathcal{P}_{\mathcal{U}_{k+1}}(X_{k+1} = i) \tag{23}$$

$$= \sum_{i=1}^{k+1} i \cdot \left( \mathcal{P}_{\mathcal{U}_{k+1}}(x_1, X_k = i - 1) + \mathcal{P}_{\mathcal{U}_{k+1}}(X_k = i) \right) \tag{24}$$

$$= \frac{1}{k+1} \cdot \sum_{i=1}^{k+1} i \cdot \mathcal{P}_{\mathcal{U}_k}(X_k = i - 1) + \sum_{i=1}^{k} i \cdot \mathcal{P}_{\mathcal{U}_{k+1}}(X_k = i) \tag{25}$$

$$= \frac{1}{k+1} \cdot \left( \sum_{i=1}^{k+1} (i-1) \cdot \mathcal{P}_{\mathcal{U}_k}(X_k = i - 1) + \sum_{i=1}^{k+1} \mathcal{P}_{\mathcal{U}_k}(X_k = i - 1) \right) + \sum_{i=1}^{k} i \cdot \mathcal{P}_{\mathcal{U}_{k+1}}(X_k = i) \tag{26}$$

$$= \frac{1}{k+1} \cdot \sum_{i=1}^{k} i \cdot \mathcal{P}_{\mathcal{U}_k}(X_k = i) + \frac{1}{k+1} \cdot \sum_{i=1}^{k} \mathcal{P}_{\mathcal{U}_k}(X_k = i) + \frac{k}{k+1} \cdot \sum_{i=1}^{k} i \cdot \mathcal{P}_{\mathcal{U}_k}(X_k = i) \tag{27}$$

$$= \frac{1}{k+1} \cdot W(k) + \frac{1}{k+1} \cdot 1 + \frac{k}{k+1} \cdot W(k) \tag{28}$$

$$= W(k) + \frac{1}{k+1} \tag{29}$$

$\mathcal{U}_k$ denotes the discrete uniform distribution over $k$ samples. Eq. (23) follows using the definition of expectation. In Eq. (24), we decompose based on the first example $x_1$ is either chosen or not. Eq. (25), Eq. (26), and Eq. (27) follow by switching the probability space from the uniform distribution $\mathcal{U}_{k+1}$ to $\mathcal{U}_k$. In Eq. (28) we apply the induction statement. Thus, we have proven that $W(n) = \sum_{i=1}^{n} \frac{1}{i}$. Since $\sum_{i=1}^{n} \frac{1}{i} \leq \mathcal{O}(\log(n+1))$, thus random-TEQ provides at the most $\mathcal{O}(\log(n+1))$ counterexamples "in the worst-case" to steer $\ell_{\text{const}}$ to the target hypothesis in $\mathcal{H}_{\text{threshold}}$. $\square$

Now, we discuss the query complexity bounds for a Max-Min query learner. The bounds achieved would be useful in analyzing the case of Axes-aligned hyperplanes in Appendix B.3.

---

[5]We use the size of the hypothesis class in the bounds.

**Min-Max query learner for Threshold functions.** To study the query complexity of Threshold functions we construct an elimination graph $G_{\text{elim}}(\mathcal{H}_{\text{threshold}}, \mathcal{U}_n)$ (see **Definition 7** [14]) which is an $(n+1) \times (n+1)$ matrix with rows and columns marked with ordered hypotheses in $\mathcal{H}_{\text{threshold}}$ and every entry corresponding to $h, h' \in \mathcal{H}_{\text{threshold}}$ is $E(h, h')$ (see Appendix B.1). First, we represent the hypothesis class $\mathcal{H}_{\text{threshold}}$ as a boolean matrix $C_{n+1}$ as

$$
\begin{array}{c}
\\ h_0 \\ h_1 \\ \vdots \\ \vdots \\ \vdots \\ h_n
\end{array}
\begin{array}{c}
\begin{array}{cccccc}
x_1 & x_2 & \cdot & \cdot & \cdot & x_n
\end{array} \\
\left(
\begin{array}{cccccc}
0 & 0 & \cdots & \cdots & \cdots & 0 \\
1 & 0 & \cdots & \cdots & \cdots & 0 \\
1 & 1 & \cdots & \cdots & \cdots & 0 \\
\vdots & \vdots & \vdots & \vdots & \vdots & 0 \\
1 & 1 & 1 & 1 & 1 & 0 \\
1 & 1 & \cdots & \cdots & \cdots & 1
\end{array}
\right)
\end{array}
$$

which has $n + 1$ rows for the hypotheses and $n$ columns for the examples. Now, we would construct the **elimination graph** corresponding to $C_{n+1}$ for uniform distribution over the examples. It is not very difficult to note that the elimination graph $G_{\text{elim}}(C_{n+1}, \mathcal{U}_n)$ has the form:

$$
\begin{array}{c}
\\ h_0 \\ h_1 \\ h_2 \\ \vdots \\ \vdots \\ h_n
\end{array}
\begin{array}{c}
\begin{array}{cccccc}
h_0 & h_1 & h_2 & h_3 & \cdots & h_n
\end{array} \\
\left(
\begin{array}{cccccc}
0 & \frac{1}{(n+1)} & \frac{3}{2(n+1)} & \frac{2}{(n+1)} & \cdots & \frac{1}{2} \\
\frac{n}{(n+1)} & 0 & \frac{2}{(n+1)} & \frac{5}{2(n+1)} & \cdots & \frac{n}{2(n-1)} - \frac{1}{(n-1)(n+1)} \\
\frac{(2n-1)}{2(n+1)} & \frac{(n-1)}{(n+1)} & 0 & \frac{3}{(n+1)} & \cdots & \frac{n}{2(n-2)} - \frac{1}{(n-2)(n+1)} \\
\vdots & \vdots & \vdots & \vdots & \vdots & \vdots \\
\vdots & \vdots & \vdots & \vdots & \vdots & \vdots \\
\frac{1}{2} & \cdot & \cdots & \cdots & \cdots & 0
\end{array}
\right)
\end{array}
$$

We note that $\ell_{\text{Max-Min}}$ picks the middle threshold function for querying based on Eq. (21). Thus, $\ell_{\text{Max-Min}}$ performs a *binary search* over the set of consistent hypotheses. Thus, in the worst-case teaching scenario $\ell_{\text{Max-Min}}$ queries at the least $\Omega\left(\log(n+1)\right)$ times. Since binary search algorithm over Threshold functions is optimal in terms of query complexity (use induction on $j$ where $n = 2^j$), thus $\ell_{\text{Max-Min}}$ is an optimal learner for Threshold functions. But we also note that even in the presence of best-TEQ the learner $\ell_{\text{Max-Min}}$ has to query at least twice for the worst-case threshold function, and hence LwEQ-TD$_{\mathcal{X}, \mathcal{H}_{\text{threshold}}, h_0}(\ell_{\text{Max-Min}}) = 2$ as shown for a constant query learner.

**Remarks.** For the hypothesis class of Threshold functions, there could be ties when finding $\arg\max_{h \in \mathcal{H}(Z_{t-1})} \min_{h' \in \mathcal{H}(Z_{t-1})\setminus\{h\}} E(h, h')$ (rhs of Eq. (21)). A simple case is when the size $n$ of the input space is 1. At time step $t = 0$, we obtain the $(2 \times 2)$ elimination graph $G_{\text{elim}}$ such that $E(h_0, h_1) = E(h_1, h_0) = \frac{1}{2}$. This observation could be generalized for any odd natural number $n$. In particular, one can show that at time step $t = 0$, both the hypotheses $h_{\frac{n-1}{2}}$ and $h_{\frac{n+1}{2}}$ maximize Eq. (21) such that $E(h_{\frac{n-1}{2}}, h_{\frac{n+1}{2}}) = E(h_{\frac{n+1}{2}}, h_{\frac{n-1}{2}}) = \frac{1}{2}$.

In the case of Threshold functions, the intuition that $\ell_{\text{Max-Min}}$ turns out to be performing binary search is based on the computation of the elimination graph $G_{\text{elim}}$, as shown above. At time step $t = 0$, it is clear that $\ell_{\text{Max-Min}}$ (using Eq. (21)) picks the middle threshold function. For time step $t > 0$, we observe that the version space is a continuous interval of threshold functions (i.e., if $h, h'$ are in the version space then every threshold in between $h$ and $h'$ is also in the version space). Hence, we can again use the computation of the elimination graph as shown for time step $t = 0$. This is the main intuition behind the query.

## B.3 Analysis for Axes-aligned Hyperplanes

In this section, we provide the proofs to the query complexity bounds for $\ell_{\text{const}}$, $\ell_{\text{global}}$, and $\ell_{\text{Max-Min}}$ under different teaching scenarios as shown in Table 2.

In the rest of the section, we refer a hypothesis $h \in \mathcal{H}_{\text{axh}}$ based on the axis of alignment $i$ and the index $j$, where $i \in [1, d+1]$, $j \in [n+1]$ such that $\forall \mathbf{x} \in \mathcal{X}$, $h(\mathbf{x}) = 1$ if $x_i \leq j$, otherwise 0.

For the sake of continuity, we redefine some of the quantities discussed in Section 4.1. We state the bounds in terms of $|\mathcal{H}_{\mathrm{axh}}| = d \cdot (n+1)$. To capture the global ranking of $\ell_{\mathrm{global}}$ we use the function $g : \mathcal{H}_{\mathrm{axh}} \to [|\mathcal{H}_{\mathrm{axh}}|]$. For a target hypothesis $h^* \in \mathcal{H}_{\mathrm{axh}}$ and ranking function $g$, the bounds for $\ell_{\mathrm{global}}$ are described using the quantities—$t_g^* := |\{h \in \mathcal{H}_{\mathrm{axh}} \mid g(h) \leq g(h^*)\}|$ (note $0 \leq t_g^* \leq d \cdot (n+1)$). We denote the set $\{h \in \mathcal{H}_{\mathrm{axh}} \mid g(h) \leq g(h^*)\}$ as $\mathcal{H}^*$. Now, we redefine the quantity $n_g^*$ as $n_g^* := \max_{m \in [1,d+1]} |\{h \in \mathcal{H}^* : h \text{ is aligned along axis } m\}|$. Note, $0 \leq n_g^* \leq n+1$.

To show the worst-case teaching bounds we don't impose any restrictions on the learners $\ell_{\mathrm{const}}$, $\ell_{\mathrm{global}}$, and $\ell_{\text{Max-Min}}$. On the other hand, our bound for best-case teaching in the case of global learners is for a specific learner $\ell_{\mathrm{global}}$ depending on the target hypothesis $h^*$. To establish the results, we state the following theorems and provide their proofs[6]: Theorem 7 states the bounds for worst-TEQ, Theorem 8 states the bounds for random-TEQ, and Theorem 9 states the bounds for best-TEQ.

**Theorem 7** (Worst-case teaching). *Consider the hypothesis class of Axes-aligned hyperplanes $\mathcal{H}_{\mathrm{axh}}$ (see Eq. (7)). In the LwEQ paradigm, the following bounds on the query complexity hold for worst-TEQ:*

1. *For a constant query learner $\ell_{\mathrm{const}}$ the query complexity is lower bounded by $\Omega(d \cdot (n+1))$.*

2. *For a global query learner $\ell_{\mathrm{global}}$ the query complexity is lower bounded by $\Omega(t_g^*)$.*

3. *For a Max-Min learner $\ell_{\text{Max-Min}}$ the query complexity is lower bounded by $\Omega(d \cdot \log(n+1))$.*

*Proof.* *Constant and Global query learners:* Consider counterexamples of the form:

$$\mathbf{x} = \Big(0, \cdots, 0, \underset{\text{k-th component}}{j}, 0, \cdots, 0\Big), \quad \mathbf{x}' = \Big(n, \cdots, n, \underset{\text{k-th component}}{j}, n, \cdots, n\Big). \tag{30}$$

$(\mathbf{x}, 1)$ or $(\mathbf{x}', 0)$ eliminates at most 1 hyperplane aligned along an axis $p \neq k$. Using these counterexamples worst-TEQ provides at least $\Omega(n+1)$ counterexamples "in the worst-case" along an axis (see Threshold functions in Appendix B.2) thus for $\mathcal{H}_{\mathrm{axh}}$ the query complexity is lower bounded by $\Omega(|\mathcal{H}_{\mathrm{axh}}|) = \Omega(d \cdot (n+1))$. Using a similar argument one achieves the lower bound of $\Omega(|\mathcal{H}^*|) = \Omega(t_g^*)$ for global query learner $\ell_{\mathrm{global}}$.

*Max-Min query learner:* Fix an arbitrary target hypothesis $h^*$ aligned along an axis $i^*$. Consider hypotheses $h, h' \in \mathcal{H}_{\mathrm{axh}}$. Assume that $h$ is aligned along axis $k$ and indexed at $i \in [1, n+1]$, whereas $h'$ is aligned along axis $k'$ and indexed at $j \in [1, n+1]$.

To prove the lower bound we show the following: $i)$ worst-TEQ "could" pick counterexamples so that less than *half* of the consistent hypotheses along an axis are eliminated in the version space, and $ii)$ "in the worst-case" $\ell_{\text{Max-Min}}$ has to query so that all the hypotheses aligned along any axis are eliminated in the final version space to locate the target hypothesis.

We note that if $\ell_{\text{Max-Min}}$ queries $h$ for equivalence such that target hypothesis is aligned along a different axis, counterexamples based on Eq. (30) ensure at most one hypothesis aligned along $p \neq k$ is eliminated as well as worst-TEQ could pick $(\mathbf{x}, 1)$ or $(\mathbf{x}', 0)$ such that at most *half* consistent hypotheses aligned along $k$ are eliminated. This completes the proof for $i)$.

Assume that $\ell_{\text{Max-Min}}$ locates the target hypothesis $h^*$ in the query protocol of Algorithm 1 so that $\mathcal{H}_{\mathrm{axh}}(Z)$ contains hypotheses aligned along an axis other than $i^*$ where $Z$ is the history of counterexamples received. We pick such a hypothesis $\hat{h} \in \mathcal{H}_{\mathrm{axh}}(Z)$. Note $\hat{h}$ is consistent with the counterexamples in $Z$. Thus if $\hat{h}$ is chosen as the "target" hypothesis, the learner $\ell_{\text{Max-Min}}$ would be fooled in querying $h^*$, and thus the query complexity increases. So, the assumption is invalidated. So, $Z$ must not have any consistent hypothesis aligned along an axis other than $i^*$. This completes the proof of $ii)$.

Using $i)$ and $ii)$ we show that $\ell_{\text{Max-Min}}$ queries at the least $\Omega(\log(n+1))$[7] times in an axis other than $i^*$. Since we know that for Threshold functions worst-TEQ provides at the least $\Omega(\log(n+1))$

---

[6]We skip a theorem for binary-TEQ as the results are self-explanatory.
[7]We use the number of hypotheses aligned along an axis, i.e. $(n+1)$.

counterexamples to a Max-Min learner $\ell_{\text{Max-Min}}$, thus even along the axis $i^*$ (one dimensional threshold functions), the Max-Min learner $\ell_{\text{Max-Min}}$ queries at least $\Omega\left(\log(n+1)\right)$ times. Hence, we have shown that the query complexity for $\ell_{\text{Max-Min}}$ is lower bounded by $\Omega\left(d \cdot \log(n+1)\right)$. $\quad\square$

In the following, we state the theorem for random-TEQ when teaching $\ell_{\text{const}}$, $\ell_{\text{global}}$, and $\ell_{\text{Max-Min}}$.

**Theorem 8** (Random teaching). *Consider the hypothesis class of Axes-aligned hyperplanes $\mathcal{H}_{\text{axh}}$ (see Eq. (7)). In the LwEQ paradigm, the following bounds on the query complexity, i.e. the number of counterexamples provided "in expectation" hold for random-TEQ:*

1. *For a constant query learner $\ell_{\text{const}}$ the query complexity is upper bounded by $\mathcal{O}\left(d \cdot \log(n+1)\right)$.*

2. *For a global query learner $\ell_{\text{global}}$ the query complexity is upper bounded by $\mathcal{O}\left(d \cdot \log n_g^*\right)$.*

3. *For a Max-Min learner $\ell_{\text{Max-Min}}$ the query complexity is upper bounded by $\mathcal{O}\left(\log(d \cdot (n+1))\right)$.*

*Proof.* First, we note that the upper bound on the query complexity for a Max-Min learner follows directly using the $\mathcal{O}\left(\log|\mathcal{H}|\right)$ (for finite hypothesis class $\mathcal{H}$) bound on the number of counterexamples in the random case as shown in [14]. So, we provide the proofs for $\ell_{\text{const}}$ and $\ell_{\text{global}}$.

*Constant query learner:* To show the upper bound we argue that, if $h \in \mathcal{H}_{\text{axh}}$ aligned along the axis $k$ (dimension), is the next hypothesis picked by the learner $\ell_{\text{const}}$ for equivalence query then the expected number of hypothesis eliminated in the current version space, say $H$, when random-TEQ provides counterexamples to $h$ is at the least *half* of the consistent hypotheses along the axis $k$ in the version space $H$. This is sufficient to yield the upper bound.

We argue for $d = 2$. Similar analysis works for $d > 2$ as the probability mass is integrated to the inputs corresponding to the case of $d = 2$. Consider a hypothesis $h' \not\equiv h \in \mathcal{H}_{\text{axh}}$. We note that when $h$ and $h'$ are aligned along the same axis, then using the analysis of Threshold functions (see Appendix B.2), it could be easily shown that "in expectation" a random counterexample to $h$ for $h'$ eliminates at least half of the hyperplanes in the version space. We denote by $\mathbf{R}_{h,h'}^k$ the expected number of consistent hyperplanes eliminated along the axis $k$ when random-TEQ provides counterexamples to $h$ for $h'$ where $h'$ is aligned along an axis other than $k$. Notice that the set of counterexamples are composed of inputs in two blocks ('+' for label 1 and '-' for label 0) as shown in 2D representation of the hypothesis class below.

$$
i \begin{pmatrix} + & + & & & \Big| & & & \\ + & + & & & \Big| & & & \\ & & & & \Big| & & & \\ & & & & \Big| & - & - & - \\ & & & & \Big| & - & - & - \end{pmatrix}
$$
$$
\phantom{i}\qquad\qquad j
$$

We compute $\mathbf{R}_{h,h'}^k$ as follows:
$$
\mathbf{R}_{h,h'}^k = \frac{(n-j) \cdot i \cdot (2n-i+1) + j \cdot (n-i) \cdot (n+i+1)}{2\left[i \cdot (n-j) + j \cdot (n-i)\right]} \tag{31}
$$

where $h$ and $h'$ are the $i$-th and $j$-th indexed hyperplanes in their respective axes. $\mathbf{R}_{h,h'}^k$ Here we show the computation when none of the hyperplanes aligned along an axis are eliminated. Change of parameters achieves the same result in the generic case. We need to show that $\mathbf{R}_{h,h'}^k \geq \frac{n+1}{2}$ irrespective of the choice of $i$ and $j$. We note that:
$$
(n-j) \cdot i \cdot (2n-i+1) + j \cdot (n-i) \cdot (n+i+1)
$$
$$
= (n-j) \cdot i \cdot (n+1) + (n-j) \cdot i \cdot (n-i) + j \cdot (n-i) \cdot (n+1) + j \cdot (n-i).i
$$
$$
\geq (n+1) \cdot i \cdot (n-j) + (n+1) \cdot j \cdot (n-i)
$$

Thus, $\mathbf{R}_{h,h'}^k \geq \frac{n+1}{2}$. This implies that if the constant query learner $\ell_{\text{const}}$ receives a counterexample to $h$ then at the least *half* of the consistent hypotheses along $k$ is eliminated from the current version space. Thus, the learner $\ell_{\text{const}}$ queries at most $\mathcal{O}\left(\log(n+1)\right)$ times to random-TEQ to either

locate the target or eliminate all the hypotheses along an axis. Since there are $d$ dimensions, thus random-TEQ provides at most $\mathcal{O}\left(d \cdot \log(n+1)\right)$ counterexamples "in expectation".

*Global query learner:* Now, we show the query complexity bound for a global query learner $\ell_{\text{global}}$ based on the quantity $n_g^*$. Using Eq. (31) we note that in expectation at least half of the consistent hypotheses are eliminated along an axis if a random counterexample is provided. Since for any axis $k$ there are at most $n_g^*$ consistent hypotheses in $\mathcal{H}^*$ thus for a global query learner $\ell_{\text{global}}$ random-TEQ provides at the most $\mathcal{O}\left(d \cdot \log n_g^*\right)$ counterexamples "in expectation". □

In the following, we state the theorem for best-TEQ when teaching $\ell_{\text{const}}$, $\ell_{\text{global}}$, and $\ell_{\text{Max-Min}}$.

**Theorem 9** (Best-case teaching). *Consider the hypothesis class of Axes-aligned hyperplanes $\mathcal{H}_{\text{axh}}$ (see Eq. (7)). In the LwEQ paradigm, the following bounds on LwEQ-TD hold for best-*TEQ*:*

1. *For a constant query learner $\ell_{\text{const}}$, LwEQ-TD($\ell_{\text{const}}$) = 2.*

2. *$\exists$ a global query learner $\ell_{\text{global}}$, LwEQ-TD($\ell_{\text{global}}$) = 1.*

3. *For a Max-Min learner $\ell_{\text{Max-Min}}$, LwEQ-TD($\ell_{\text{Max-Min}}$) = 2.*

*Proof. Constant query learner:* We note that two counterexamples corresponding to the opposite sides of the target hyperplane are sufficient to fix the target hyperplane in the version space. Assume $h^*$ be the target hypothesis aligned along an arbitrary axis $k$ and indexed at $i \in [1, n+1]$, i.e., $\forall \mathbf{x} \in \mathcal{X}$

$$h^*(\mathbf{x}) = \begin{cases} 1 & \text{if } x_k \leq i \\ 0 & \text{otherwise} \end{cases}$$

Consider $\mathbf{x}', \mathbf{x}'' \in \mathcal{X}$ such that $x_k' = i$, $x_k' = i+1$, and all the other coordinates are same. Note, $h^*(\mathbf{x}') = 1$ and $h^*(\mathbf{x}'') = 0$. But then any other hypothesis $h' \neq h^* \in \mathcal{H}_{\text{axh}}$ classifies both $x', x''$ either 0 or 1. Thus, $\mathcal{H}_{\text{axh}}(\{(\mathbf{x}', 1), (\mathbf{x}'', 0)\}) = \{h^*\}$. Since the learner $\ell_{\text{const}}$ arbitraily picks a hypothesis in the version space to query LwEQ-TD($\ell_{\text{const}}$) = 2.

*Global query learner:* Consider the global query learner $\ell_{\text{global}}$ with the global ranking $g : \mathcal{H}_{\text{axh}} \to [|\mathcal{H}_{\text{axh}}|]$ as follows: hypothesis which classifies the least number of inputs as negative, i.e. 0 is ranked highest (thus picked if consistent in the version space). Alternatively, for all $h, h' \in \mathcal{H}_{\text{axh}}$, $g(h) \leq g(h')$ if

$$|\{\mathbf{x} : \mathbf{x} \in \mathcal{X}, h(\mathbf{x}) = 1\}| \leq |\{\mathbf{x} : \mathbf{x} \in \mathcal{X}, h'(\mathbf{x}) = 1\}|$$

Now, if the target hypothesis is $h^*$ s.t. it is aligned along the axis $i^* \in [1, d+1]$ and indexed at $j \in [n+1]$, then best-TEQ provides a counterexample $\mathbf{x} := \left(0, \cdots, 0, \underset{i^*\text{-th component}}{j}, 0, \cdots, 0\right)$ with label 1. Note, $\ell_{\text{global}}$ picks $h^*$ in the version space $\mathcal{H}_{\text{axh}}(\{(\mathbf{x}, 1)\})$. Hence, the result follows.

*Max-Min query learner:* We note that similar to a constant query learner, LwEQ-TD($\ell_{\text{Max-Min}}$) = 2 as a Max-Min query learner picks the next hypothesis prefixed by the pairwise computation of the expected number of elimination of hypotheses when an equivalence query is posed with hypothesis $h \in \mathcal{H}_{\text{axh}}$ and target hypothesis $h' \in \mathcal{H}_{\text{axh}}$ (see Appendix B.1). As noted in Appendix B.2, $\ell_{\text{Max-Min}}$ would perform a binary search to locate a target hypothesis but then the analysis for a constant query learner $\ell_{\text{const}}$ would apply. Hence, the result follows. □

# C  Lower and Upper Bounds on Query Complexity (Section **4.2**)

In this appendix, we provide the proofs for the bounds shown in Table 3 on the query complexity for different hypothesis classes as discussed in Section 4.2. We divide the appendix in the following way: Appendix C.1 provides the proof to Theorem 1, Appendix C.2 provides the proof to Theorem 2, and Appendix C.3 provides the proof to Theorem 3.

First, we highlight an important theorem from [14] that connects the VC dimension of a hypothesis class and its query complexity for random-TEQ. Our proof for the lower bound on the query complexity in the random teaching scenario for any query learner involves the computation of VC dimension of the underlying hypothesis class.

**Theorem 10** (Theorem 25 [14])**.** *If $\mathcal{H}$ is a hypothesis class of VC-dimension d, then any randomized learning algorithm to learn $\mathcal{H}$ must use at least an expected $\Omega(d)$ equivalence queries with random counterexamples for some target hypothesis.*

Now, we provide the proofs to the theorems in the subsequent appendices.

## C.1  Axes-aligned Hyperplanes

In the following, we prove Theorem 1 which establishes lower bounds and upper bound on different teaching scenarios for Axes-aligned hyperplanes. Since the size of $\mathcal{H}_{\mathrm{axh}}$ is $d \cdot (n+1)$ thus any learner queries at least $\Omega(d \cdot (n+1))$ times to binary-TEQ.

*Proof of Theorem 1.* *Worst-case teaching:* Consider a powerful learner $\ell$ which performs the following procedure: first finds the dimension (axis) $i$ along which target hypothesis $h^*$ is aligned and then perform a binary search along the axis $i$. We show that $\ell$ is optimal and has a lower bound of $\Omega(d + \log(n+1))$ on the query complexity for worst-TEQ "in the worst-case".

For the sake of contradiction, assume $\ell'$ learns the worst-case hypothesis in less than $2d + \log(n+1)$ counterexamples. Consider counterexamples of the form:

$$\mathbf{x} = \Big(0, \cdots, 0, \underset{\text{k-th component}}{j}, 0, \cdots, 0\Big), \quad \mathbf{x}' = \Big(n, \cdots, n, \underset{\text{k-th component}}{j}, n, \cdots, n\Big).$$

Notice $\mathcal{H}_{\mathrm{axh}}(\{(\mathbf{x}, 1)\})$ or $\mathcal{H}_{\mathrm{axh}}(\{(\mathbf{x}', 0)\})$ contains at least one hypothesis aligned along axes $p \neq k$. Thus, worst-TEQ provides at the least $\Omega(d)$ counterexamples, say, $\mathcal{X}_d \subset \mathcal{X}$ such that $\forall h' \in \mathcal{H}_{\mathrm{axh}}(\mathcal{X}_d)$, $h'$ is aligned along the axis $i$ (i.e. for target hypothesis). First, we argue that worst-TEQ could choose to provide at least one counterexample of the form discussed above along each axis. Consider the case when $d = 2$. Assume that the target hypothesis is aligned along axis-1 but the learner queries a hypothesis aligned along axis-2. "In the worst-case", $(\mathbf{x}, 1)$ and $(\mathbf{x}', 0)$ (where $x_2 = x_2' = j$) forms valid labeled counterexamples to equivalence queries for hypotheses aligned along axis-2 and consistent with target hypothesis which is aligned along axis-1. Using a similar analysis, we generalize this to arbitrary dimension $d > 2$. This implies, any learner queries at least one hypothesis along each axis.

On the other hand, for any learner, worst-TEQ provides at the least $\Omega(\log(n+1))$ counterexamples "in the worst-case" (see Appendix B.2). Hence, we show that $\ell$ is optimal and worst-TEQ provides at the least $\Omega(d + \log(n+1))$ counterexamples "in the worst-case".

*Random teaching:* The key to showing the lower bound of $\Omega(\log d)$ on the query complexity for random-TEQ, i.e number of counterexamples "in expectation", for any learner involves showing a lower bound on the VC dimension of Axes-aligned hyperplanes as defined in Definition 4.1.

Denote the set of combinations of $\log d$ objects as $\mathbb{C}^d$, where $\mathbb{C}_i^d$ represents $i$-th combination in $\mathbb{C}^d$. Consider the set of $\log d$ inputs $H_{\mathrm{vc}} := \{\mathbf{x}_1, \mathbf{x}_2, \cdots, \mathbf{x}_{\log d}\}$ such that for every component $i \in [1, d+1]$ the $(\log d)$-tuple $(x_{1,i}, x_{2,i}, \cdots, x_{\log d,i}) = (1, 2, \cdots, \log d)_{\mathbb{C}_i^d}$, i.e. the tuple $(1, 2, \cdots, \log d)$ sets index not in $\mathbb{C}_i^d$ as $n$ else other keeps the values. Now, consider a $(\log d)$-tuple $\mathbf{b}$ of boolean values. We note that there exists an axis $i$ and index $j$ such that the corresponding hypothesis $h'$ (i.e. $\forall \mathbf{x} \in \mathcal{X}$, $h(\mathbf{x}) = 1$ if $x_i \leq j$, otherwise 0) classifies $H_{\mathrm{vc}}$ as $\mathbf{b}$ (component-wise). This holds because for all $m \in [1, (\log d)+1]$ such that $\mathbf{b}_m = 1$ (or $\mathbf{b}_m = 0$) ($m$-th boolean value in $\mathbf{b}$) $x_{m,i} \leq j$ (or $x_{m,i} > j$) for the $(\log d)$-tuple $(x_{1,i}, x_{2,i}, \cdots, x_{\log d,i})$. This implies that $H_{\mathrm{vc}}$ is

*shattered* by $\mathcal{H}_{\mathrm{axh}}$. Thus, $\mathrm{VCD}(\mathcal{H}_{\mathrm{axh}}) = \Omega\left(\log d\right)$, which directly gives a lower bound on the query complexity for random-TEQ, i.e. on the number of counterexamples provided "in expectation".

*Best-case teaching:* In Theorem 9 (see Appendix B.3) we show there exists a global query learner $\ell_{\mathrm{global}}$ such that the query complexity for best-TEQ is 1, i.e. $\mathrm{LwEQ\text{-}TD}_{\mathcal{X}, \mathcal{H}_{\mathrm{axh}}, h_0} = 1$.  □

## C.2 Monotone Monomials

Now, we proof Theorem 2 which establishes lower bounds and upper bound on query complexity in different teaching scenarios for Monotone monomials. For the sake of clarity, we rewrite Eq. (8) here

$$\mathcal{H}_{\mathrm{mono}} := \{h \mid \exists\ \{i_1, i_2, \cdots, i_k\} \subseteq \{1, 2, \cdots, n\},\ \mathrm{s.t.}\ \forall \mathbf{x} \in \mathcal{X},\ \ h(\mathbf{x}) = x_{i_1} \wedge \cdots \wedge x_{i_k}\}.$$

We note that $|\mathcal{H}_{\mathrm{mono}}| = 2^n$. Thus, any learner queries at least $\Omega\left(2^n\right)$ times to binary-TEQ. Furthermore, we note that any hypothesis $h \in \mathcal{H}_{\mathrm{mono}}$ represented as $h(\mathbf{x}) := x_{i_1} \wedge x_{i_2} \wedge \cdots \wedge x_{i_k}$ for some $\{i_1, i_2, \cdots, i_k\} \subseteq \{1, 2, \cdots, n\}$ is "identified" with an input $\mathbf{x}' \in \mathcal{X}$ such that for all $m \in \{i_1, i_2, \cdots, i_k\}, x'_m = 1$ otherwise 0.

*Proof of Theorem 2.* *Worst-case teaching:* Consider a global learner $\ell$ which ranks hypothesis based on the number of "dependent" literals, i.e. for all $h, h' \in \mathcal{H}_{\mathrm{mono}}$, $g(h) \le g(h')$ (where $g$ is the global ranking for $\ell$) if $h$ has at least as many dependent literals as $h'$. We show that $\ell$ is optimal and has a lower bound of $\Omega\left(n\right)$ on the query complexity for worst-TEQ "in the worst-case".

First, we show that $\ell$ asks at least $\Omega\left(n\right)$ queries to worst-TEQ "in the worst-case". We use the representation of a monomial in terms of its set of literals. Fix a starting hypothesis $h_0 := \mathrm{mono}\left(j_1, j_2, \cdots, j_m\right)$. Consider the following illustration for query protocol of Algorithm 1:

$$\underbrace{\mathrm{mono}\left(j_1, j_2, \cdots, j_m\right)}_{\text{starting monomial } h_0} \xrightarrow{t=0} \underbrace{\mathrm{mono}\left(1, 2, \cdots, n\right)}_{\ell \text{ queries for equivalence}} \xrightarrow{t=1} \cdots \xrightarrow{t=t'} \underbrace{\mathrm{mono}\left(i_1, i_2, \cdots, i_k\right)}_{\text{target monomial}}$$

Notice that the counterexample $(\mathbf{x}_1, y_1)$ to $\mathrm{mono}\left(1, 2, \cdots, n\right)$ has to be a positive example (i.e, $h^*(\mathbf{x}_1) = y_1 = 1$) as $\{i_1, i_2, \cdots, i_k\} \subset \{1, 2, \cdots, n\}$. Also, worst-TEQ picks $\mathbf{x}_1$ such that the number of 1's in the string is $n - 1$ to maximize the query complexity. Thus, it is straightforward that at each time step worst-TEQ provides a counterexample such that the subset of literals of the corresponding hypothesis (or monomial) is a superset of $\{i_1, i_2, \cdots, i_k\}$ but has size 1 less than the subset size for the current hypothesis (monomial). Essentially, the set $\{1, 2, \cdots, n\} \setminus \{i_1, i_2, \cdots, i_k\}$ is eliminated before the learner $\ell$ queries $h^*$ for equivalence. This implies that $\ell$ performs at least $\Omega\left(n\right)$ queries to identify a target "in the worst-case" if worst-TEQ provides counterexamples.

Now, we argue there doesn't exist a learner $\ell'$ with a better query complexity. For the sake of contradiction, assume that some learner $\ell'$ achieves a query complexity less than $n$ for worst-TEQ. Note that $\ell'$ can't query the target hypothesis, at some time step, such that there exists a monomial $\mathrm{mono}\left(i_1, i_2, \cdots, i_k, i_{k+1}\right)$ where $\{i_1, i_2, \cdots, i_k\} \subsetneq \{i_1, i_2, \cdots, i_k, i_{k+1}\}$ in the current version space. This holds because, if $Z'$ is the current history of counterexamples provided by $\ell'$, then $Z'$ is also a valid set of counterexamples for $\mathrm{mono}\{i_1, i_2, \cdots, i_k, i_{k+1}\}$ as a target hypothesis. Thus, worst-TEQ fools the learner $\ell'$ in this case. So, $\ell'$ queries monomials in a way to eliminate at least all the supersets of $\{i_1, i_2, \cdots, i_k\}$. But if a monomial corresponding to a superset, say $\mathrm{mono}\{j_1, j_2, \cdots, j_{k'}\}$, is queried for equivalence, worst-TEQ could follow the strategy described for the learner $\ell$. So, monomials with the set of literals, say $S'$ such that $\{i_1, i_2, \cdots, i_k\} \subseteq S' \subseteq \{j_1, j_2, \cdots, j_{k'}\}$ remain consistent. Since "in the worst-case" $\ell'$ has to query successively to reduce the size of the set of literals $S'$ of the queried monomial, $\ell'$ performs at least $\Omega\left(n\right)$ queries. So, our assumption on the query complexity for worst-TEQ teaching $\ell'$ is wrong, thus $\ell$ is optimal.

*Random-case teaching:* The key to showing the lower bound of $\Omega\left(n\right)$ on the query complexity for random-TEQ, i.e number of counterexamples "in expectation", for any learner involves showing a lower bound on the VC dimension of the hypothesis class of Monotone monomials as defined in Definition 4.2. Consider the set $M_{\mathrm{vc}}$ defined as:

$$M_{\mathrm{vc}} := \{\mathbf{x} \in \mathcal{X} : \exists\ !i \in [1, n+1]\ \mathrm{s.t.}\ x_i = 0\}$$

We show that $M_{\mathrm{vc}}$ is *shattered* by $\mathcal{H}_{\mathrm{mono}}$. We note that for all $h \not\equiv h' \in \mathcal{H}_{\mathrm{mono}}$, $h(M_{\mathrm{vc}}) \ne {}^{8} h'(M_{\mathrm{vc}})$. To show this consider a literal $v_{i'}$ present in $h$ but not in $h'$. Now, for $\mathbf{x} \in M_{\mathrm{vc}}$ such that $x_{i'} = 0$,

---

[8] We define $h$ on a subset of $\mathcal{X}$ as a tuple of boolean values.

$h(\mathbf{x}) = 0$ but $h'(\mathbf{x}) = 1$. Since $|\mathcal{H}_{\mathrm{mono}}| = 2^n$ and $M_{\mathrm{vc}} = n$, we have shown that $M_{\mathrm{vc}}$ is shattered by $\mathcal{H}_{\mathrm{mono}}$, leading to VCD($\mathcal{H}_{\mathrm{mono}}$) = $n$. Thus, for any optimal learner the lower bound on query complexity for random-TEQ, i.e. on the number of counterexamples provided "in expectation" is $\Omega(n)$.

*Best-case teaching:* Consider the global query learner $\ell_{\mathrm{global}}$ with the global ranking $g : \mathcal{H}_{\mathrm{mono}} \rightarrow [|\mathcal{H}_{\mathrm{mono}}|]$ as follows: ranks hypothesis based on the number of "dependent" literals, i.e. if $h, h' \in \mathcal{H}_{\mathrm{mono}}$ then $g(h) \leq g(h')$ if $h$ has at least as many dependent literals as $h'$. At the beginning, $\ell_{\mathrm{global}}$ queries $h(\mathbf{x}) \equiv x_1 \wedge x_1 \wedge \cdots \wedge x_n$. Then the optimal teacher best-TEQ provides $(\mathbf{x}', 1)$ as a counterexample where $\mathbf{x}'$ identifies $h^*$ as discussed above. Now, the highest ranked monomial in $\mathcal{H}_{\mathrm{mono}}(\{(\mathbf{x}', 1)\})$ is $h^*$ which the learner $\ell_{\mathrm{global}}$ queries for equivalence. Since best-TEQ provides exactly 1 counterexample the worst-case hypothesis in $\mathcal{H}_{\mathrm{mono}}$, thus LwEQ-TD$_{\mathcal{X}, \mathcal{H}_{\mathrm{mono}}, h_0}(\ell_{\mathrm{global}}) = 1$ (for a fixed $h_0 \in \mathcal{H}_{\mathrm{mono}}$). Hence, the result follows. $\qquad \square$

## C.3 Orthogonal Rectangles

Now, we prove Theorem 3 which establishes lower bounds and upper bound on query complexity in different teaching scenarios for Orthogonal rectangles. We note that the size of $\mathcal{H}_{\mathrm{rec}}$ is of the order $\Omega\left((n \cdot (n+1))^d\right)$, thus any learner queries at least $\Omega\left((n \cdot (n+1))^d\right)$ times to binary-TEQ.

*Proof of Theorem 3. Worst-case teaching:* Similar to the case of worst-case teaching for Axes-aligned hyperplanes (see Definition 4.1), we show that any optimal learner has a lower bound of $\Omega(d \cdot \log n)$ on the query complexity for worst-TEQ "in the worst-case".

Consider a target hypothesis $h^* \in \mathcal{H}_{\mathrm{rec}}$ such that for all $j \in [1, d+1], a_j = 0$. The choice of $b_j$ is shown later. Now, consider counterexamples of the form:

$$\mathbf{x} = \Big(0, \cdots, 0, \underset{\text{k-th component}}{m}, 0, \cdots, 0\Big).$$

Notice that labeled counterexamples $(\mathbf{x}, 0)$ or $(\mathbf{x}, 1)$ to $h' \in \mathcal{H}_{\mathrm{rec}}$ doesn't affect the choice of $b_j$ (doesn't get fixed for consistent hypotheses in the version space) for all $j\ (\neq k) \in [1, d+1]$. This implies that worst-TEQ could provide counterexamples such that any learner has to query hypotheses for equivalence along each axis $i \in [1, d+1]$. Using the analysis of Threshold functions (see Appendix B.2) we know that the worst-TEQ provides at least $\Omega(\log(n+1))$ counterexamples to any learner along an axis. but, we could pick worst-case $b_j$'s for a target hypothesis $h^*$ such that worst-TEQ provides at least $\Omega(\log(n+1))$ counterexamples in every axis, implying that for any learner worst-TEQ provides at the least $\Omega(d \cdot \log(n+1))$ counterexamples "in the worst-case".

*Random teaching:* The key to showing the lower bound of $\Omega(d)$ on the query complexity for random-TEQ i.e number of counterexamples "in expectation", for any learner involves showing a lower bound on the VC dimension of the hypothesis class of Orthogonal rectangles as defined in Definition 4.3. Consider the set $O_{\mathrm{vc}}$ defined as:

$$O_{\mathrm{vc}} := \{\mathbf{x} \in \mathcal{X} : \exists i \in [1, d+1], \text{ s.t. } x_i = \lfloor n/2 \rfloor \text{ and } \forall j \neq i, x_j = 0\}$$

We note that $|O_{\mathrm{vc}}| = d$. Now, we show that $O_{\mathrm{vc}}$ is shattered by $\mathcal{H}_{\mathrm{rec}}$. Notice that for any $d$-tuple $\mathbf{b}$ of boolean values, one can fix the choices of $\{a_j, b_j\}_{j \in [1, d+1]} \subset [n+1]$ to obtain a hypothesis $h \in \mathcal{H}_{\mathrm{rec}}$ where $a_j = 0$, and $b_j = \lfloor n/2 \rfloor$ if $\mathbf{b}_i = 1$ else $b_j = \lfloor n/2 \rfloor - 1$. This gives $h(O_{\mathrm{vc}}) = \mathbf{b}$. So, we have shown that $O_{\mathrm{vc}}$ is *shattered* by $\mathcal{H}_{\mathrm{rec}}$, which implies VCD($\mathcal{H}_{\mathrm{rec}}$) = $\Omega(d)$. Thus, for any optimal learner the lower bound on the query complexity for random-TEQ, i.e. on the number of counterexamples provided "in expectation" is $\Omega(d)$.

*Best-case teaching:* We show that there is a global query learner $\ell_{\mathrm{global}}$ which achieves the required query complexity bound of 2. In order to establish this, we note for fixed $h_0, \mathcal{X}, \mathcal{H}$, LwEQ-TD$_{\mathcal{X}, \mathcal{H}, h_0}(\Sigma_{\sigma_{\mathrm{global}}})$ = RTD($\mathcal{H}$) (see Theorem 4, Section 5) for the family of ranking functions $\Sigma_{\sigma_{\mathrm{global}}}$ (see Section 5) which induce global query learners. We need to show that RTD($\mathcal{H}_{\mathrm{rec}}$) = 2. [25] showed that the TD (teaching dimension) notion of preference-based teaching dimension (PBTD) of a finite hypothesis class $\mathcal{H}$ is the same as the RTD of the class $\mathcal{H}$ (see **Corollary 9** [25]). On the other hand, [61] showed that PBTD($\mathcal{H}_{\mathrm{rec}}$) = 2 (see **Example 1** [61]). Thus, there exists a global query learner $\ell_{\mathrm{global}}$ such that LwEQ-TD$_{\mathcal{X}, \mathcal{H}_{\mathrm{rec}}, h_0}(\ell_{\mathrm{global}}) = 2$. $\qquad \square$

## D  Teaching Dimensions for the LwEQ and LfS Paradigms (Section 5)

In this appendix, we prove our main results for Section 5, i.e., Theorem 4 and Theorem 5, which establish connections of LwEQ-TD in the learning-with-equivalence-queries (LwEQ) paradigm to existing notions of TD in the learning-from-samples (LfS) paradigm. We divide the appendix based on the connections established for learner types induced by a specific family of ranking functions.

First, we note that the framework of ranking functions (see Eq. (10), Section 5) leads to the definition of a preference function as discussed in [29]. Thus, a ranking function $\sigma$ could be, alternatively, used as a preference function to define LfS-TD$_{\mathcal{X},\mathcal{H},h_0}(\sigma)$ (for fixed $\mathcal{X}, \mathcal{H}, h_0$) for a learner induced by a preference function $\sigma$ as shown in [29] in the LfS paradigm. We illustrate this as follows:

$$\sigma \text{ (ranking function)} \quad\rightsquigarrow\quad \sigma \text{ (preference function)}$$
$$\uparrow \qquad\qquad\qquad\qquad\qquad\qquad \downarrow$$
$$\text{Eq. (10)} \qquad\qquad\qquad\qquad \textbf{Protocol 1 } [29]$$
$$\text{\small(LwEQ paradigm)} \qquad\qquad\qquad \text{\small(LfS paradigm)}$$

With this understanding, we would interchangeably talk about teaching settings in the LfS paradigm for learner models (in the LfS paradigm) induced by a framework of ranking functions. Our proof technique uses the unification of teaching settings in the LfS paradigm [29].

### D.1  LwEQ Learners with Ranking Functions Independent of $Z_{t-1}$ and $h_{t-1}$

For fixed $\mathcal{X}, \mathcal{H}, h_0$, we prove Theorem 4 establishing connections for LwEQ-TD$_{\mathcal{X},\mathcal{H},h_0}(\Sigma_{\text{const}})$ and LwEQ-TD$_{\mathcal{X},\mathcal{H},h_0}(\Sigma_{\text{global}})$.

*Proof of Theorem 4.* First, we show the connection for the family of *constant* learners $\Sigma_{\text{const}}$. We note that any sequential teaching set $Z \subseteq \mathcal{Z}$ in the LfS paradigm is permutation invariant. For arbitrary $h_0$ and $h^*$, if $Z := \{z_1, z_2, \cdots, z_l\}$ is the minimal teaching set for teaching in the LfS paradigm then any permutation of $z_1, z_2, \cdots, z_l$ is also a teaching set ($\Sigma_{\text{const}}$ leads to a batched teaching setting in the LfS paradigm). Since all the hypotheses are preferred equally, they are eliminated in the teaching protocol at some time step. This implies that in the query protocol of Algorithm 1 in the LwEQ paradigm, at any step $t$, the teacher could pick $z_t \in Z$ such that $z_t$ forms a counterexample to the queried hypothesis $h_t \in \ell_{\sigma_{\text{const}}}(Z_{t-1}, h_{t-1})$. Thus, the teacher could always permute $Z$ to respond as counterexamples in Algorithm 1 so as to steer the learner from $h_0$ to $h^*$. Since $h_0$ and $h^*$ are arbitrary, thus LwEQ-TD$_{\mathcal{X},\mathcal{H},h_0}(\sigma_{\text{const}}) \leq$ LfS-TD$_{\mathcal{X},\mathcal{H},h_0}(\sigma_{\text{const}})$.

Because any history of counterexamples $Z \subseteq \mathcal{Z}$ used in the query protocol of Algorithm 1 to steer the learner from $h_0$ to $h^*$, forms a solution for teaching in the LfS paradigm, we have LwEQ-TD$_{\mathcal{X},\mathcal{H},h_0}(\sigma_{\text{const}}) =$ LfS-TD$_{\mathcal{X},\mathcal{H},h_0}(\sigma_{\text{const}})$. Now, we note the following:

$$\min_{\sigma_{\text{const}} \in \Sigma_{\text{const}}} \text{LwEQ-TD}_{\mathcal{X},\mathcal{H},h_0}(\sigma_{\text{const}}) = \min_{\sigma_{\text{const}} \in \Sigma_{\text{const}}} \text{LfS-TD}_{\mathcal{X},\mathcal{H},h_0}(\sigma_{\text{const}}) \tag{32}$$

$$\implies \text{LwEQ-TD}_{\mathcal{X},\mathcal{H},h_0}(\Sigma_{\text{const}}) = \text{LfS-TD}_{\mathcal{X},\mathcal{H},h_0}(\Sigma_{\text{const}}) \tag{33}$$

Eq. (32) is straightforward since for any $\sigma_{\text{const}} \in \Sigma_{\text{const}}$, LwEQ-TD$_{\mathcal{X},\mathcal{H},h_0}(\sigma_{\text{const}}) =$ LfS-TD$_{\mathcal{X},\mathcal{H},h_0}(\sigma_{\text{const}})$. Eq. (33) follows using the definition of LwEQ-TD for a family of query functions $\Sigma_{\text{const}}$ and the definition of LfS-TD for a family of preference functions $\Sigma_{\text{const}}$ (see [29]). Now, we note that using [29] (see Theorem 1) shows that LfS-TD$_{\mathcal{X},\mathcal{H},h_0}(\Sigma_{\text{const}}) =$ wc-TD$(\mathcal{H})$, which implies LwEQ-TD$_{\mathcal{X},\mathcal{H},h_0}(\Sigma_{\text{const}}) =$ wc-TD$(\mathcal{H})$.

Now, for a *global* ranking function $\sigma_{\text{global}} \in \Sigma_{\text{global}}$ we note that any hypothesis which is not ranked strictly over the target hypothesis $h^*$ is never queried for equivalence. Thus, $H^* = \{h \in \mathcal{H} \mid g(h) \leq g(h^*)\}$ where $g$ is the global function for $\sigma_{\text{global}}$, needs to be eliminated both in the query protocol in the LwEQ paradigm and the teaching protocol in the LfS paradigm. Using similar ideas as discussed above for *constant* query learners, we note that LwEQ-TD$_{\mathcal{X},\mathcal{H},h_0}(\sigma_{\text{global}}) =$ LfS-TD$_{\mathcal{X},\mathcal{H},h_0}(\sigma_{\text{global}})$, and hence LwEQ-TD$_{\mathcal{X},\mathcal{H},h_0}(\Sigma_{\text{global}}) =$ LfS-TD$_{\mathcal{X},\mathcal{H},h_0}(\Sigma_{\text{global}})$. Since LfS-TD$_{\mathcal{X},\mathcal{H},h_0}(\Sigma_{\text{global}}) =$ RTD$(\mathcal{H})$ (Theorem 1, [29]), we show LwEQ-TD$_{\mathcal{X},\mathcal{H},h_0}(\Sigma_{\text{global}}) =$ RTD$(\mathcal{H})$. $\qquad\square$

**Remark 1.** When $\mathcal{H}$ is singleton, LwEQ-TD$_{\mathcal{X},\mathcal{H},h_0}(\ell) = 0$ as per the query complexity defined in Eq. (5) for a query learner $\ell$. In the proof above, we implicitly assumed that $\mathcal{H}$ is not singleton.

**Remark 2.** Note that if there exists a target hypothesis $h^* \in \mathcal{H}$ for which $\ell(\emptyset, h_0) = \{h^*\}$, then LwEQ-TD$_{\mathcal{X},\mathcal{H},h_0}(\ell, h^*) = 0$ as per the query complexity defined in Eq. (4); however, the teaching complexity in the LfS paradigm (see **Protocol 1** [29]) leads to LfS-TD$_{\mathcal{X},\mathcal{H},h_0}(\ell, h^*) = 1$. Since we are interested in analyzing $\max_{h^* \in \mathcal{H}}$ LwEQ-TD$_{\mathcal{X},\mathcal{H},h_0}(\ell, h^*)$, the proofs focus on $h^* \in \mathcal{H}$ for which LwEQ-TD$_{\mathcal{X},\mathcal{H},h_0}(\ell, h^*) > 0$.

## D.2 LwEQ Learners with Ranking Functions Dependent on $Z_{t-1}$ and/or $h_{t-1}$

For fixed $\mathcal{X}, \mathcal{H}, h_0$, we prove Theorem 5 establishing connections for LwEQ-TD$_{\mathcal{X},\mathcal{H},h_0}(\Sigma_{\mathrm{gvs}})$, LwEQ-TD$_{\mathcal{X},\mathcal{H},h_0}(\Sigma_{\mathrm{local}})$, and LwEQ-TD$_{\mathcal{X},\mathcal{H},h_0}(\Sigma_{\mathrm{lvs}})$.

*Proof of Theorem 5. Connection for $\Sigma_{\mathrm{gvs}}^{\mathrm{CF}}$.* Fix an arbitrary global version space learner characterized with a ranking function $\sigma_{\mathrm{gvs}} \in \Sigma_{\mathrm{gvs}}^{\mathrm{CF}}$. Furthermore, assume that $g$ identifies $\sigma_{\mathrm{gvs}}$ in $\Sigma_{\mathrm{gvs}}^{\mathrm{CF}}$ (see Eq. (15), Section 5.2).

First, we note that any history of counterexamples $Z \subseteq \mathcal{Z}$ used in the query protocol of Algorithm 1 to steer the learner from $h_0$ to $h^*$, forms a solution for teaching in the LfS paradigm for the preference function $\sigma_{\mathrm{gvs}}$ (see **Protocol 1** [29]). Denote $|Z|$ by $k$ and $Z_k := Z$. Now, we know that $\ell_{\sigma_{\mathrm{gvs}}}(Z_k, h_k) = \arg\min_{h' \in \mathcal{H}(\{Z_k\})} \sigma_{\mathrm{gvs}}(h'; \mathcal{H}(\{Z_k\}), h_k) = \{h^*\}$. But then for all $h$, $\arg\min_{h' \in \mathcal{H}(\{Z_k\})} \sigma(h'; \mathcal{H}(\{Z_k\}), h) = \arg\min_{h' \in \mathcal{H}(\{Z_k\})} g(h'; \mathcal{H}(\{Z_k\}))$. This implies that $Z$ forms a teaching set in the teaching protocol as discussed in [29] in the LfS paradigm. Since $h^*$ is arbitrarily picked, we have LfS-TD$_{\mathcal{X},\mathcal{H},h_0}(\sigma_{\mathrm{gvs}}) \leq$ LwEQ-TD$_{\mathcal{X},\mathcal{H},h_0}(\sigma_{\mathrm{gvs}})$.

Assume $Z' := \{z_1, z_2, \cdots, z_l\}$ is the minimal teaching set for teaching in the LfS paradigm. The optimal teacher (LwEQ paradigm) could pick a permutation of $Z'$ to provide as counterexamples in the LwEQ paradigm. At time $t$, if $Z_{t-1}$ is the current history of counterexamples (assuming $Z_{t-1} \subseteq Z'$) and $h_{t-1}$ is the current hypothesis, then in the query protocol of Algorithm 1 the learner picks the next hypothesis as follows

$$h_t \in \arg\min_{h' \in \mathcal{H}(Z_{t-1})} \sigma_{\mathrm{gvs}}(h'; \mathcal{H}(Z_{t-1}), h_{t-1}) = \arg\min_{h' \in \mathcal{H}(Z_{t-1})} g(h', \mathcal{H}(Z_{t-1}))$$

Notice that either *i)* $h_t = h^*$ or *ii)* $\exists z \in Z'$ s.t. $z$ is a counterexample to $h_t$. Assume that neither *i)* nor *ii)* hold. Then $\nexists z \in Z'$ such that $z$ is a counterexample to $h_t$. This contradicts the collusion-freeness of $\sigma_{\mathrm{gvs}}$ (see Definition 5.1) because

$$\arg\min_{h' \in \mathcal{H}(Z_{t-1})} \sigma_{\mathrm{gvs}}(h'; \mathcal{H}(Z_{t-1}), h_{t-1}) = \{h_t\},$$

but

$$\arg\min_{h' \in \mathcal{H}(Z_{t-1} \cup (Z' \setminus Z_{t-1}))} \sigma_{\mathrm{gvs}}(h'; \mathcal{H}(Z'), h_t) = \{h^*\}$$

which contradicts the fact that $h_t$ is consistent with $Z' \setminus Z_{t-1}$. Hence, LwEQ-TD$_{\mathcal{X},\mathcal{H},h_0}(\sigma_{\mathrm{gvs}}) \leq$ LfS-TD$_{\mathcal{X},\mathcal{H},h_0}(\sigma_{\mathrm{gvs}})$. Since we picked $\sigma_{\mathrm{gvs}}$ arbitrarily, we have

$$\min_{\sigma_{\mathrm{gvs}} \in \Sigma_{\mathrm{gvs}}^{\mathrm{CF}}} \text{LwEQ-TD}_{\mathcal{X},\mathcal{H},h_0}(\sigma_{\mathrm{gvs}}) = \min_{\sigma_{\mathrm{gvs}} \in \Sigma_{\mathrm{gvs}}^{\mathrm{CF}}} \text{LfS-TD}_{\mathcal{X},\mathcal{H},h_0}(\sigma_{\mathrm{gvs}})$$

$$\overset{\text{(Eq. (6), Section 3)}}{\Longrightarrow} \text{LwEQ-TD}_{\mathcal{X},\mathcal{H},h_0}(\Sigma_{\mathrm{gvs}}^{\mathrm{CF}}) = \text{LfS-TD}_{\mathcal{X},\mathcal{H},h_0}(\Sigma_{\mathrm{gvs}}^{\mathrm{CF}})$$

$$\overset{\text{(Theorem 1 [29])}}{\Longrightarrow} \text{LwEQ-TD}_{\mathcal{X},\mathcal{H},h_0}(\Sigma_{\mathrm{gvs}}^{\mathrm{CF}}) = \text{NCTD}(\mathcal{H})$$

The last equation yields the desired result.

*Connection for $\Sigma_{\mathrm{local}}^{\mathrm{CF}}$.* First, we prove the lower bound on LwEQ-TD$_{\mathcal{X},\mathcal{H},h_0}(\Sigma_{\mathrm{local}}^{\mathrm{CF}})$ as stated in Eq. (19). To show the bound, we note that LfS-TD$_{\mathcal{X},\mathcal{H},h_0}(\Sigma_{\mathrm{local}}^{\mathrm{CF}}) :=$ local-PBTD$_{\mathcal{X},\mathcal{H},h_0}$ (see [29]).

Fix $h_0$ and $h^*$ as the starting and target hypotheses. Now, fix the best ranking function $\sigma_{\mathrm{local}}^* := \arg\min_{\sigma_{\mathrm{local}} \in \Sigma_{\mathrm{local}}^{\mathrm{CF}}}$ LwEQ-TD$_{\mathcal{X},\mathcal{H},h_0}(\sigma_{\mathrm{local}})$, i.e. the best ranking function which minimizes the

LwEQ-TD for the family of ranking functions $\Sigma_{\text{local}}^{\text{CF}}$. Assume the optimal teacher provides the following counterexamples to the learner $\ell_{\sigma_{\text{local}}^*}$ in the query protocol of Algorithm 1:

$$h_0 \xrightarrow{\emptyset} h_1 \xrightarrow{Z_1} h_2 \cdots \xrightarrow{Z_{k+1}} h^* \tag{34}$$

where $Z_i := Z_{i-1} \cup \{z_{i-1}\}$ for the counterexample $z_{i-1}$ to $h_{i-1}$. We show that there exists a choice of a local ranking function $\sigma_{\text{local}}' \in \Sigma_{\text{local}}^{\text{CF}}$ such that the history of counterexamples $Z_{k+1}$ provided by the optimal teacher in Algorithm 1 as shown above forms a sequence of teaching examples for a learner characterized with the local ranking $\sigma_{\text{local}}'$ in the LfS paradigm (see **Protocol 1** [29]).

Define $\sigma_{\text{local}}'$ as follows: for all $h' \in \mathcal{H}(\{z_1\})$, $\sigma_{\text{local}}'(h'; \mathcal{H}(\{z_1\}), h_0) = \sigma_{\text{local}}^*(h'; \mathcal{H}(\{z_1\}), h_1)$ (see Eq. (34)) where $z_1$ is the counterexample provided by the optimal teacher to $h_1$ (hypothesis queried for equivalence with the current history of examples $\emptyset$, and the current hypothesis $h_0$) for the learner characterized by the ranking function $\sigma_{\text{local}}^*$ in the query protocol of Algorithm 1 as described above; otherwise $\sigma_{\text{local}}' \equiv \sigma_{\text{local}}^*$. Thus, $\arg\min_{h' \in \mathcal{H}(\{z_1\})} \sigma_{\text{local}}'(h'; \mathcal{H}(\{z_1\}), h_0) = \{h_2\}$ (see Eq. (34)). Since $h^*$ is arbitrary, thus $\sigma_{\text{local}}'$ could be defined over all the triplets $(h^*, z_1, h_2)$, i.e. for any choice of target $h^*$, as shown in Eq. (34). Notice that $\sigma_{\text{local}}'$ is a valid local ranking function as $\sigma_{\text{local}}^*$ is a local ranking function. Similarly, the collusion-freeness of $\sigma_{\text{local}}'$ follows from its definition and the collusion-freeness of $\sigma_{\text{local}}^*$.

For the learner characterized by $\sigma_{\text{local}}'$ in the LfS paradigm, the optimal teacher could provide the counterexamples $Z_{k+1}$ sequentially for teaching in **Protocol 1** [29]. This is illustrated as follows:

$$h_0 \xrightarrow{Z_1} h_2 \xrightarrow{Z_2} h_3 \cdots \xrightarrow{Z_{k+1}} h^* \tag{35}$$

Upon receiving the examples $z_i$, the learner picks the next hypothesis $h_{i+1}$ as shown above. Thus, $Z_{k+1}$ forms a teaching set in the LfS paradigm. This implies that LfS-TD$_{\mathcal{X},\mathcal{H},h_0}(\sigma_{\text{local}}', h^*) \leq |Z_{k+1}|$. Since $h^*$ is arbitrarily chosen, thus we have LfS-TD$_{\mathcal{X},\mathcal{H},h_0}(\sigma_{\text{local}}') \leq$ LwEQ-TD$_{\mathcal{X},\mathcal{H},h_0}(\sigma_{\text{local}}^*)$. But this implies

$$\min_{\sigma_{\text{local}} \in \Sigma_{\text{local}}^{\text{CF}}} \text{LfS-TD}_{\mathcal{X},\mathcal{H},h_0}(\sigma_{\text{local}}) \leq \text{LfS-TD}_{\mathcal{X},\mathcal{H},h_0}(\sigma_{\text{local}}') \leq \text{LwEQ-TD}_{\mathcal{X},\mathcal{H},h_0}(\sigma_{\text{local}}^*)$$

$$\overset{(\sigma_{\text{local}}^* \text{ minimizes LwEQ-TD}(\cdot))}{\implies} \text{LfS-TD}_{\mathcal{X},\mathcal{H},h_0}(\Sigma_{\text{local}}^{\text{CF}}) \leq \text{LwEQ-TD}_{\mathcal{X},\mathcal{H},h_0}(\Sigma_{\text{local}}^{\text{CF}})$$

Thus, from the last equation we have local-PBTD$_{\mathcal{X},\mathcal{H},h_0} \leq$ LwEQ-TD$_{\mathcal{X},\mathcal{H},h_0}(\Sigma_{\text{local}}^{\text{CF}})$.

The upper bound on LwEQ-TD$_{\mathcal{X},\mathcal{H},h_0}(\sigma_{\text{local}})$ follows by noting that LwEQ-TD$_{\mathcal{X},\mathcal{H},h_0}(\sigma_{\text{local}}) =$ wc-TD$(\mathcal{H})$ for a ranking function $\sigma_{\text{const}} \in \Sigma_{\text{const}}$ (as shown in Theorem 4) and that *constant* query learner could be adversarial compared to the *local* learners.

We would show the inequality using induction. Fix a local ranking function $\sigma_{\text{local}} \in \Sigma_{\text{local}}^{\text{CF}}$, and a constant ranking function $\sigma_{\text{const}} \in \Sigma_{\text{const}}$. Now, consider arbitrary $h_0$ and $h^*$ as the starting and target hypotheses respectively. The induction statement is on the worst-case optimal teaching sequence size for steering the learner from a starting to a target hypothesis when the learner is characterized with a constant ranking function. The inequality holds for $D_{\sigma_{\text{const}}}(Z \subseteq \mathcal{Z}, h_0, h^*) = 0$. By definition, for any $Z \subseteq \mathcal{Z}, h_0, h^*$, we have $D_{\sigma_{\text{local}}}(Z, h_0, h^*) \leq D_{\sigma_{\text{const}}}(Z, h_0, h^*) = 0$ as $\ell_{\sigma_{\text{local}}}(Z, h_0) = \ell_{\sigma_{\text{const}}}(Z, h_0) = \{h^*\}$. Following the induction statement, assume that $D_{\sigma_{\text{local}}}(Z \subseteq \mathcal{Z}, h_0, h^*) \leq D_{\sigma_{\text{const}}}(Z \subseteq \mathcal{Z}, h_0, h^*)$ whenever $D_{\sigma_{\text{const}}}(Z \subseteq \mathcal{Z}, h_0, h^*) \leq k$. Now, we need to show the inequality when $D_{\sigma_{\text{const}}}(\emptyset, h_0, h^*) = k+1$ (similar argument holds for $Z \subseteq \mathcal{Z}$). We unfold the recursion for $D_{\sigma_{\text{const}}}$:

$$D_{\sigma_{\text{const}}}(\emptyset, h_0, h^*) = 1 + \max_{h' \in \ell_{\sigma_{\text{const}}}(\emptyset, h_0)} \min_{z: h'(x_z) \neq y_z} D_{\sigma_{\text{const}}}(\{z\}, h', h^*) \tag{36}$$

We note that for all $h' \in \ell_{\sigma_{\text{const}}}(\emptyset, h_0)$,

$$\min_{z: h'(x_z) \neq y_z} D_{\sigma_{\text{const}}}(\{z\}, h', h^*) \geq \min_{z: h'(x_z) \neq y_z} D_{\sigma_{\text{local}}}(\{z\}, h', h^*).$$

Since $\ell_{\sigma_{\text{local}}}(\emptyset, h_0) \subseteq \ell_{\sigma_{\text{const}}}(\emptyset, h_0)$ we have:

$$\max_{h' \in \ell_{\sigma_{\text{const}}}(\emptyset, h_0)} \min_{z: h'(x_z) \neq y_z} D_{\sigma_{\text{const}}}(\{z\}, h', h^*) \geq \max_{h' \in \ell_{\sigma_{\text{local}}}(\emptyset, h_0)} \min_{z: h'(x_z) \neq y_z} D_{\sigma_{\text{local}}}(\{z\}, h', h^*)$$

Plugging this into Eq. (36) implies

$$D_{\sigma_{\mathrm{const}}}(\emptyset, h_0, h^*) \geq D_{\sigma_{\mathrm{local}}}(\emptyset, h_0, h^*)$$

Since $h^*$ is arbitrary we get LwEQ-TD$_{\mathcal{X},\mathcal{H},h_0}(\sigma_{\mathrm{const}}) \geq$ LwEQ-TD$_{\mathcal{X},\mathcal{H},h_0}(\sigma_{\mathrm{local}})$ for arbitrary $\sigma_{\mathrm{const}}$ and $\sigma_{\mathrm{local}}$. Using the definition of LwEQ-TD for a family of ranking functions (Eq. (6) and Section 5), we get

$$\min_{\sigma_{\mathrm{const}} \in \Sigma_{\mathrm{const}}} \mathrm{LwEQ\text{-}TD}_{\mathcal{X},\mathcal{H},h_0}(\sigma_{\mathrm{const}}) \geq \min_{\sigma_{\mathrm{local}} \in \Sigma^{\mathrm{CF}}_{\mathrm{local}}} \mathrm{LwEQ\text{-}TD}_{\mathcal{X},\mathcal{H},h_0}(\sigma_{\mathrm{local}})$$

Hence we get LwEQ-TD$_{\mathcal{X},\mathcal{H},h_0}(\Sigma^{\mathrm{CF}}_{\mathrm{local}}) \leq$ wc-TD$(\mathcal{H})$ using Theorem 4.

*Connection for* $\underline{\Sigma^{\mathrm{CF}}_{\mathrm{lvs}}}$. Using similar arguments as used to show the bounds in Eq. (19) for the family of local ranking functions $\Sigma^{\mathrm{CF}}_{\mathrm{local}}$, the bounds in Eq. (20) follow as well. $\qquad\square$

**Is there a gap between lvs-PBTD$_{\mathcal{X},\mathcal{H},h_0}$ and LwEQ-TD$_{\mathcal{X},\mathcal{H},h_0}(\Sigma^{\mathrm{CF}}_{\mathrm{lvs}})$?** In Theorem 5, we showed that lvs-PBTD$_{\mathcal{X},\mathcal{H},h_0} \leq$ LwEQ-TD$_{\mathcal{X},\mathcal{H},h_0}(\Sigma^{\mathrm{CF}}_{\mathrm{lvs}})$. An interesting research question could be to understand the gap between lvs-PBTD$_{\mathcal{X},\mathcal{H},h_0}$ and LwEQ-TD$_{\mathcal{X},\mathcal{H},h_0}(\Sigma^{\mathrm{CF}}_{\mathrm{lvs}})$. We answer this question partially and leave the unresolved part for future work. We present a problem instance that suggests that there could be a strict gap between lvs-PBTD$_{\mathcal{X},\mathcal{H},h_0}$ and LwEQ-TD$_{\mathcal{X},\mathcal{H},h_0}(\Sigma^{\mathrm{CF}}_{\mathrm{lvs}})$.

**Theorem 11.** *For learners whose query function is induced by a ranking function dependent on $Z_{t-1}$ and/or $h_{t-1}$, there exists a problem instance of $\mathcal{X}, \mathcal{H}, h_0, \sigma_{\mathrm{lvs}} \in \Sigma^{\mathrm{CF}}_{\mathrm{lvs}}$, such that*

$$\mathrm{LfS\text{-}TD}_{\mathcal{X},\mathcal{H},h_0}(\sigma_{\mathrm{lvs}}) \ll \mathrm{LwEQ\text{-}TD}_{\mathcal{X},\mathcal{H},h_0}(\sigma_{\mathrm{lvs}})$$

*i.e.,* LfS-TD$_{\mathcal{X},\mathcal{H},h_0}(\sigma_{\mathrm{lvs}})$ *is much lower than* LwEQ-TD$_{\mathcal{X},\mathcal{H},h_0}(\sigma_{\mathrm{lvs}})$.

*Proof.* Consider a $d$-dimensional lattice of finite size $n$ (isomorphic to a cube in $\mathbb{R}^d$ of length $n$ with positive integer coordinates). $\mathcal{H}$ and $\mathcal{X}$ correspond to the nodes of the lattice. The hypothesis $h_v$ corresponding to node $v$ is identified as one which classifies $v' \neq v$ as positive and $v$ as negative. We consider learners characterized with the ranking function $\sigma_{\mathrm{lvs}} \in \Sigma^{\mathrm{CF}}_{\mathrm{lvs}}$ such that it moves to a close-by hypothesis measured via $\ell_1$ (Manhattan) distance. It is not very difficult to note that LwEQ-TD$_{\mathcal{X},\mathcal{H},h_0}(\sigma_{\mathrm{lvs}})$ is at least $\Omega(n^d)$.

Now, we argue that LfS-TD$_{\mathcal{X},\mathcal{H},h_0}(\sigma_{\mathrm{lvs}})$ is upper bounded by $\mathcal{O}(n \cdot d^2)$. Consider the hypotheses[9] $h_0 = \big(\underbrace{0,0,0,0,\cdots 0,0,0,0}_{d}\big)$ and $h^* = \big(\underbrace{n,n,n,n,\cdots n,n,n,n}_{d}\big)$. The teacher could try to steer the learner in the following sequential form:

$$h_0 \dashrightarrow h_1 \dashrightarrow h_2 \dashrightarrow \cdots \dashrightarrow h_i \dashrightarrow \cdots \dashrightarrow h_{d-1} \dashrightarrow h_d = h^*$$

where $h_i = \big(\underbrace{n,n,n,n,\cdots,n,n,n}_{i},0,0,\cdots,0\big)$. We show that with $\mathcal{O}(n \cdot d)$ teaching examples the learner could be taught to move from $h_i$ to $h_{i+1}$ for any $i \in [n]$. Notice that the learner prefers to stay on $h_v$ for a node unless $v$ is provided as a positive example to steer it away. Thus, the teacher first provides $\Big\{(0, i_1, i_2, i_3, \cdots, i_{d-1}) \Big| \sum_{j=1}^{d-1} i_j \leq 1 \text{ s.t. } \exists\,!k\ i_k \neq 0 \Big\}$. Then, the teacher provides $\{(a, 0, 0, \cdots, 0)\}_{a=1}^{n}$ in the sequential order. Therefore, in $\mathcal{O}(n \cdot d)$ examples the learner moves from $h_0$ to $h_1$. Using a similar strategy, the teacher requires at most $\mathcal{O}(n \cdot d^2)$ examples ($\mathcal{O}(n \cdot d)$ for each dimension) to steer the learner to the target $h^*$. It can be easily verified that the same holds for any $h_0$ and $h^*$ and thus LfS-TD$_{\mathcal{X},\mathcal{H},h_0}(\sigma_{\mathrm{lvs}})$ is $\mathcal{O}(n \cdot d^2)$. Hence, for the hypothesis class $\mathcal{H}$ of a lattice in $\mathbb{R}^d$ of size $n$ with the input space $\mathcal{X} := \{1, 2, \cdots, n\}^d$ we get,

$$\mathcal{O}(n \cdot d^2) = \mathrm{LfS\text{-}TD}_{\mathcal{X},\mathcal{H},h_0}(\sigma_{\mathrm{lvs}}) \ll \mathrm{LwEQ\text{-}TD}_{\mathcal{X},\mathcal{H},h_0}(\sigma_{\mathrm{lvs}}) = \Omega(n^d).$$

$\qquad\square$

---

[9]We represent each hypothesis by the node that identifies it.