# OpenReview forum: "Teaching via Best-Case Counterexamples in the Learning-with-Equivalence-Queries Paradigm"
_NeurIPS.cc/2021/Conference — NeurIPS 2021 Poster_

### Official Review · Reviewer_AepY · 2021-07-13

**Rating:** 8
**Confidence:** 2

**Summary:**

This work studies the sample complexity of machine teaching finite hypothesis classes with equivalence queries (EQ), a model in which the learner queries hypotheses in the concept class directly, and the “teacher” responds “yes” or “no” along with a counter-example with the goal of eventually teaching the learner some ground truth concept.

There are many formalizations of this model in the literature. This work mainly focuses on a natural regime of the problem in which the teacher is “helpful,” always choosing the best possible counter-example to help the learner, and the learner is adversarial, though generally restricted under some fixed “learning rule,” (e.g. they must always pick a hypothesis consistent with previous counter-examples).

The authors study a notion of query complexity they call “Learning with Equivalence Queries Teaching Dimension” (LwEQ-TD) that is closely related to other forms of TD used in machine teaching. They show that for a variety of fundamental concept classes (axis-aligned halfspaces, axis-aligned rectangles, and monotone monomials), a helpful teacher can always teach even the most adversarial learner in O(1) examples. On the other hand, they give lower bounds on the sample complexity when the teacher gives random or adversarial samples that scales with the size of the class.

Finally, the authors give an in depth comparison between LwEQ-TD and previous notions of machine teaching in the membership query model. Indeed, for a very natural class of adversarial learner, they prove that LwEQ-TD is in fact exactly the standard teaching dimension (along with a host of other connections).


**Main Review:**

This work introduces an original model of machine teaching that is both combinatorially natural, and well motivated in real life. Previous work studying equivalence queries either used a “random” or “adversarial” teacher. To my knowledge, this is the first work to study the scenario of a helpful teacher and a (lightly) adversarial student in the EQ model.

Overall, the results in this work should be of broad interest to the learning community. They prove non-trivial gaps between learning with a helpful and random teacher for several fundamental classes, and establish strong connections with numerous standard notions of teaching complexity such as teaching dimension and recursive teaching dimension. While none of these results alone are of great technical interest, they establish important connections to standard results in an original (and natural) model, which is a valuable contribution.

Overall the paper is well written, though some details appear to be missing (see below). The proofs I checked were technically sound.

A few minor comments for the authors:

Typos: “less benign”  should be “less adversarial” or “more benign,” “contant” should be “constant”

Should $h’$ be over should be $Z \cup \{z\}$ in Eq 3?

It’d be nice to have some justification as to why the learning function can’t depend on $h_{j}$ for $j<t-1$

Reference to learning rules says Section 2, but probably should be app:B.1

Is the learner rule stated somewhere in Table 3?

Learning rules as an abstraction (rather than a specific rule) are never formally defined.

The various definitions of LfS-TD compared to need to be defined somewhere. If possible in the main body, at least informally. Definitely in an appendix.

Edit: I thank the authors for their response and clarifications.

**Time Spent Reviewing:**

7

---

> ### Author Response · Authors · 2021-08-06
> **Response to Reviewer AepY**
>
> Thank you for carefully reviewing our paper! Please see below our responses to your comments.
>
> -----
> **1. Typo: “less benign”**
> - Thanks for pointing out the typo. Indeed, we meant to write “more benign” instead of “less benign”.
>
> **2. Should $h’$ be over ($Z \cup z$) in Eq 3?**
> - We interpret the reviewer’s comment to include ($Z \cup z$) under the max operator in Eq. 3.  In the query protocol in Algorithm 1 (page 4, line 139),  the learner picks a hypothesis first, and then a counterexample is provided by the teacher. Hence, the learner’s next hypothesis $h’$ depends on $Z$ under the max operator in Eq. 3, instead of ($Z \cup z$).
>
> **3. It’d be nice to have some justification as to why the learning function can’t depend on $h_j$  for $j < t-1$.**
> - Thanks a lot for the suggestion. In Section 5, we studied specific families of learner query functions with dependence on  $\mathcal{H}(Z_{t-1})$ (Eq. 15), $h_{t-1}$ (Eq. 16), and $\mathcal{H}(Z_{t-1})$, $h_{t-1}$ (Eq. 17). These families are well-studied in the LfS paradigm and allowed us to connect our results to the existing notions of LfS-TD through these families.
> - In comparison to Section 5, our generic query function in Section 2 (Eq. 2) captures a broader family with the dependency on the complete history of examples  $Z_{t-1}$ received and $h_{t-1}$. In principle, we could further extend this to have a dependency on the complete sequence of queries (i.e., sequence of hypotheses $h_0$, $h_1$, ..., $h_{t-1}$) made by the learner. It will be interesting to establish new notions of TD for these powerful query functions in future works.
>
> **4. Is the learner rule stated somewhere in Table 3?**
> - We will add more information about the learner rules in the table caption for clarity. Currently, this is discussed in Section 4.2 (lines 223-230) and we will add more details in the final revision.
> - For the setting of three teachers (binary-TEQ, worst-TEQ, random-TEQ), the lower bounds in Table 3 are established for learners characterized with the optimal query function specific to the hypothesis classes of interest. We have discussed these optimal query functions in detail in Appendix C corresponding to each hypothesis class.
> - For the setting of best-case teacher (best-TEQ), the upper bounds in Table 3 are established for a learner with global query function (lines 140-147 in Section 2; lines 202-208 Section 4.1; Appendix B.1).
>
> **5. Other minor comments**
> - We will fix all other minor comments. As the reviewer suggested, we will add informal definitions of LfS-TD in the main paper, as well as more detailed definitions in the appendix.
>
> -----
> We hope that our responses can help address your concerns. If you have any other comments or feedback, please let us know! We will be happy to provide further responses.  We are looking forward to hearing back from you!

---

### Official Review · Reviewer_ce98 · 2021-07-16

**Rating:** 7
**Confidence:** 3

**Summary:**

This work studied the problem of learning from equivalence queries (EQ) alone where counterexamples are selected by a helpful teacher. The paper characterizes the teaching complexity for different classes and relates the new notion (LwEQ-TD) to different notions in the literature. LwEQ-TD showed significant savings in the complexity of teaching when compared to other teaching notions in some hypothesis classes.

**Limitations And Societal Impact:**

This work is theoretical

**Main Review:**

This work proposes a new teaching notion called LwEQ-TD for learning from equivalence queries (EQ) only where counterexamples are carefully selected by a teacher. The main strength of the work is that it goes beyond the classical paradigm of *learning from samples* teaching notions (defined as LfS-TD) and characterizes LwEQ-TD when having different learners and loss functions.

The paper also discusses how LwEQ-TD is related to different notions in LfS-TD such as TD/RTD/ PBTD given a holistic view on different teaching models and notions, which is highly needed with the increase interest in the theories and applications of machine teaching. However, the paper is very condense in its content, proofs, and theorems. Indeed, the work provides new insights and results from a learning theory perspective albeit their practical implications are not very clear. Moreover, learning from EQ alone is not practical in many applications, this needs to be motivated in the context of LwEQ-TD as the cognitive burden of answering one EQ is arguably higher than other types of queries.



**Time Spent Reviewing:**

3

---

> ### Author Response · Authors · 2021-08-06
> **Response to Reviewer ce98**
>
> Thank you for carefully reviewing our paper! Please see below our responses to your comments.
>
> -----
> **1. Regarding practical applications of learning from EQ**
> - LwEQ paradigm has been employed for several important problems of practical significance, including designing program synthesis tools (e.g., oracle-guided synthesis [6], counterexamples-guided synthesis [10]), data augmentation [11], and learning structured hypothesis classes (e.g., automata learning [12]). In recent years, the above-mentioned problem settings are becoming increasingly popular in the context of educational applications, e.g., tutoring systems for block-based programming and data science (e.g., Microsoft’s CodeHunt application and the Automata Tutor). In the context of programming, the learner’s current program represents the EQ query to which an automated teacher can then provide a counterexample. In this problem setting, equivalence queries are more natural than membership queries.
>
> **2. The cognitive burden of answering one EQ**
> - We agree with the reviewer that an EQ query is, in general, more complex than a membership query. It would be very interesting to understand the complexity of asking and answering different types of queries in different application domains.
> - However, in the context of the above-mentioned applications with EQ queries, one typically implements the teacher via a verification tool that provides counterexamples to the queried hypotheses. In this context, the teacher is basically a computational tool and hence the cognitive burden of answering is less concerning.
>
> -----
> We will add a discussion about the above points in the updated version of the paper. We hope that our responses can help address your concerns. If you have any other comments or feedback, please let us know! We will be happy to provide further responses.  We are looking forward to hearing back from you!

---

### Official Review · Reviewer_iHcM · 2021-07-16

**Rating:** 6
**Confidence:** 3

**Summary:**

The paper characterises the sample complexity of teaching in the learning with equivalence queries(LwEQ) model. The teachers can be
1) worst, 2) average and 3) optimal. The number of samples required for the learner to reach the target hypothesis is intuitively the least in the best(3) case. This work draws a connection to the learning from samples model, specifically to the notion of teaching dimension.

**Limitations And Societal Impact:**

Yes, the authors do discuss the relevant limitations.

**Main Review:**

The paper is well written and quite readable. The definitions and the concepts are laid out cleanly.


Weakness:
1) The conjecture in the conclusion (regarding DFAs) needs more discussion. What is the intuition behind the suggested bound?
2) The paper cites a number of works that study teaching complexity(TD) in the learning-from-samples(LfS) setting.  Although this shows the importance of TD in the LfS setting, it does not say much about the importance of TD in the LwEQ setting. It would be interesting to see a more thorough discussion about the same.
3) The authors chose 3 hypothesis classes, and in all of them, the teaching complexity is in {1,2}. This does support the intuitive point of the paper that optimal teachers are powerful. However, it would be interesting to see some more complex hypothesis classes with non-trivial lower bounds on the teaching complexity. For example, as written in the conclusion, the problem of determining TD for CFGs and DFA/NFAs is quite intriguing.

Minor Comments:
1) (line 36) Do you mean a more benign teacher?

**Time Spent Reviewing:**

3

---

> ### Author Response · Authors · 2021-08-06
> **Response to Reviewer iHcM**
>
> Thank you for carefully reviewing our paper! Please see below our responses to your comments.
>
> -----
> **1. Importance of TD in the LwEQ paradigm**
> -  We will expand on the discussion in the paper. It is true that TD has been studied extensively in the LfS paradigm. However, in recent years, there is a growing interest in understanding TD in more interactive settings (e.g., teaching via explanations, teaching an active learner, teaching via demonstrations). We view LwEQ as a powerful learning paradigm in many natural application domains, as detailed in the next paragraph. Similar to the LfS paradigm, understanding TD in such contexts will not only inform the power and limitations of different types of (counter-) examples but also inspire the design of better teaching and learning algorithms.
> - In terms of application domains, the LwEQ paradigm has been extensively studied in many problem settings, including designing program synthesis tools, data augmentation, and learning structured hypothesis classes. In recent years, the above-mentioned problem settings are also becoming increasingly popular in the context of educational applications, e.g., tutoring systems for block-based programming and data science.
> - On the technical side, our work is also inspired by the recent results from [14] where it has been shown that random counterexamples can bring significant (e.g., exponential for certain classes) reduction in query complexity in comparison to worst-case counterexamples. This led us to investigate the power of best-case counterexamples in comparison to random-case and worst-case counterexamples.
>
> **2. Determining TD for CFGs and DFA/NFAs is quite intriguing; The conjecture in the conclusion (regarding DFAs) needs more discussion**
> -  Thanks for the comments. To avoid confusion for the readers, we will pose the suggested bound as an open problem instead of stating it as a conjecture.
> -  Determining LwEQ-TD for DFAs and more complex classes is definitely a very interesting problem to study. As we stated in the conclusions, the complexity for DFAs is exponential in $s$ (where $s$ is the state size) for the worst-case counterexamples [13]; only in a recent work from 2017 [14], it has been shown that the random counterexamples can achieve the complexity of $\Theta(s \log s)$. Our results on the simpler hypothesis classes show a significant reduction in the query complexity from worst-case to random counterexamples; they further suggest that the query complexity of the best-case could be substantially lower than the random-case.
> - One possible direction to make progress on this question is to study the query complexity when providing more structured counterexamples (e.g., by picking minimal length counterexamples as considered in [12]). This would also allow one to establish an upper bound for best-case counterexamples. It is indeed intriguing if $\Theta(s \log s)$ can be improved with best-case counterexamples. We hope that our work on LwEQ-TD can stir interest in this problem of establishing best-case teaching complexity for complex structured classes.
>
> **3. The authors chose 3 hypothesis classes, and in all of them, the teaching complexity is in {$1,2$}.**
> - First, we want to point out that when considering $\ell_{\textrm{const}}$ learner for three hypothesis classes, the teaching complexity is not always constant. In fact, for Monotone monomials the complexity is $\min (r+1; n)$ (where $r$ is the number of relevant variables) and for Orthogonal rectangles is ($2 + 2d$) (see reference [18]).
> - We would also like to point out that in Table 3, these upper bounds for best-case teacher (best-TEQ) are established for a learner with a global query function $\ell_{\textrm{global}}$. These bounds turn out to be constant and much lower than the bounds for $\ell_{\textrm{const}}$.
> - Moreover, Theorem 5 (Section 5.2, lines 340-343) shows that if we consider the learner's query function induced by a ranking function dependent on $\mathcal{H}(Z_{t−1})$ and $h_{t−1}$, then the LwEQ-TD notion turns out to be weaker compared to the corresponding notion of LfS-TD. More concretely, in Appendix D.2 (page 27), we show a problem instance ($\mathcal{X}, \mathcal{H}$, $h_0$, $\sigma_{\textrm{lvs}}$) using $d$-dimensional lattice of finite size $n$ for which lvs-PBTD($\sigma_{\textrm{lvs}}$) is $\mathcal{O}(n \cdot d^2)$ whereas LwEQ-TD($\sigma_{\textrm{lvs}}$) is $\Omega(n^d)$.
>
> **4. (line 36) Do you mean a more benign teacher?**
> - Thanks for pointing out the typo. Yes, we meant to write “more benign”.
>
> -----
> We hope that our responses can help address your concerns. If you have any other comments or feedback, please let us know! We will be happy to provide further responses.  We are looking forward to hearing back from you!

---

> > ### Comment · Reviewer_iHcM · 2021-08-11
> > **Response to Author Response**
> >
> > Thanks a lot for the in-depth response. I think the overall response clarified some critical points and hence I will raise my score from 5 to 6.

---

### Official Review · Reviewer_SpHM · 2021-07-18

**Rating:** 7
**Confidence:** 3

**Summary:**

This paper studies the equivalence query learning model where the teacher presents counterexamples if the learner proposes an incorrect hypothesis.  Three types of teachers are considered: (1) the "worst-case teacher" that presents adversarial counterexamples, (2) the "random teacher" that presents random counterexamples and (3) the "best-case teacher" that presents counterexamples designed to guide the learner towards the correct hypothesis in as few learning steps as possible.  Further, the query learner is equipped with a preference function on which the choice of its query depends (if its current hypothesis is incorrect).

**Ethical Concerns:**

None.

**Limitations And Societal Impact:**

Yes.

**Main Review:**

The study of equivalence query learning with helpful teachers seems to be a natural follow-up to previous work on algorithmic teaching and I think the present work is a good starting point.

Comment:

- Supplementary, page 17, line 642: Why does $\ell_{max-min}$ pick the middle threshold function for querying?  I think this point needs more justification.  I tested this observation on some values, specifically for $n = 20$ and $n = 10$, and the claim seems to be almost correct except that the all-zero concept or all-one concept maximize $E(h,h')$ when the current version space contains these two concepts.

Minor comments/suggestions:


- Page 6, Definition 4.1: Is the input space {${1,...,n}$}$^d$ or {${1,...,n+1}$}$^d$?  In the definition, $x$ is a point in this space with coordinates up to $n+1$, not $n$?


- Page 8, line 327: collude -> colluding

- Page 10, reference [17]: Missing space between 'V' and 'N'.

- Supplementary, page 17, matrix between 638 and 639: Why is the subscript of h in the last row $T$ and not $n+1$?  Similarly for the matrix between lines 641 and 642.

**Time Spent Reviewing:**

16

---

> ### Author Response · Authors · 2021-08-06
> **Response to Reviewer SpHM**
>
> Thank you for carefully reviewing our paper! Please see below our responses to your comments.
>
> -----
> **1. Why does $ℓ_{\textrm{Max}−\textrm{Min}}$ pick the middle threshold function for querying?**
> - Thank you for carefully checking the supplementary material! To clarify why $ℓ_{\textrm{Max}−\textrm{Min}}$ picks the middle threshold function for querying, we will update the paper with a more detailed discussion and a visual illustration for a fixed size $n$ of the input space.
> - In the case of Threshold functions (Appendix B.2), the intuition that $ℓ_{\textrm{Max}−\textrm{Min}}$ turns out to be performing binary search is based on the computation of the elimination graph $G_{\textrm{elim}}$, as shown in between the lines 641-642 (page 17). At time  step $t = 0$, it is clear that $ℓ_{\textrm{Max}−\textrm{Min}}$ (using Eq. 21, page 15) picks the middle threshold function. For time step $t > 0$, we observe that the version space is a continuous interval of threshold functions (i.e., if $h$, $h'$ are in the version space then every threshold in between $h$ and $h'$ is also in the version space). Hence, we can again use the computation of the elimination graph as shown for time step $t = 0$. This is the main intuition behind the query.
>
> **2. The all-zero concept or all-one concept maximizes $E(h, h')$ when the current version space contains these two concepts.**
> - We are interpreting the reviewer’s comments as follows: Either the all-zero concept or all-one concept maximizes Eq. 21 (Appendix B.1, page 15) when the current version space contains these two concepts. Please let us know if our interpretation is wrong.
> - The aforementioned scenario would not arise at any time step $t$ if the size $n$ of the input space for the hypothesis class of Threshold functions (see Definition B.1, page 15) is greater than $1$. At time step $t = 0$, both the all-zero concept and all-one concept are in the version space but none would maximize Eq. 21 (see elimination graph in lines 641-642, page 17). At time step $t = 1$, by the design of the hypothesis class of Threshold functions (see Definition B.1, page 15), one of them gets eliminated from the version space when the teacher provides a counterexample to the middle threshold function (see our answer to the first question).
>
> **3. Regarding Page 6, Definition 4.1**
> - Thanks for pointing out the mistake. The input space is \{$1,2,...,n$\}$^d$ and each coordinate of a point $x \in \mathcal{X}$ should be up to $n$.
>
> **4. Supplementary, page 17, matrix between 638 and 639. Why is the subscript of $h$ in the last row of the matrix $T$  and not $n+1$?**
> - Thanks for pointing out the mistake. We will replace $T$ with $n+1$.
>
> **5. Other minor comments**
> - Thanks a lot for the minor comments and suggestions; we will carefully fix them in the final version of the paper.
>
> -----
> We hope that our responses can help address your concerns. If you have any other comments or feedback, please let us know! We will be happy to provide further responses.  We are looking forward to hearing back from you!

---

> > ### Comment · Reviewer_SpHM · 2021-08-07
> > **Reply to Author Response**
> >
> > Thank you very much for the clarification. I had miscalculated and have now verified that $\ell_{Max-Min}$ indeed picks the middle threshold function for querying.  Perhaps one could also mention how ties are dealt with, e.g. broken arbitrarily etc. when finding $\arg \max_{h \in \mathcal{H}(Z_{t-1})} \min_{h' \in \mathcal{H}(Z_{t-1}) \setminus \{h\}} E(h,h')$.  (For the class of threshold functions, I suppose ties do not occur?)  Also, in Equation (21) should one be taking $\arg$ of the right-hand expression.
> >
> > I think the overall response to the reviews was great and will raise my score to 7.

---

> > > ### Author Response · Authors · 2021-08-09
> > > **Response to Reviewer SpHM**
> > >
> > > Thank you for carefully reading our responses and increasing the score! Please see below our response to your new comments.
> > >
> > > -----
> > > **1. How are ties dealt with by the learner $ℓ_{\textrm{Max}−\textrm{Min}}$? For the class of threshold functions, I suppose ties do not occur?**
> > >
> > > - The ties are broken arbitrarily when finding $\arg \max_{h \in \mathcal{H}(Z_{t-1})} \min_{h' \in \mathcal{H}(Z_{t-1})\setminus \{h\}} E(h,h')$.
> > > - For the hypothesis class of Threshold functions, there could be ties when finding $\arg \max_{h \in \mathcal{H}(Z_{t-1})} \min_{h' \in \mathcal{H}(Z_{t-1})\setminus \{h\}} E(h,h')$. A simple case is when the size $n$ of the input space is $1$. At time step $t = 0$, we obtain the $(2\times 2)$ elimination graph $G_{\textrm{elim}}$ (see lines 641-642, page 17) such that $E(h_0,h_1) = E(h_1,h_0) = \frac{1}{2}$. This observation could be generalized for any odd natural number $n$. In particular, one can show that at time step $t = 0$, both the hypotheses $h_{\frac{n-1}{2}}$ and $h_{\frac{n+1}{2}}$ maximize Eq. 21 such that $E(h_{\frac{n-1}{2}},h_{\frac{n+1}{2}}) = E(h_{\frac{n+1}{2}}, h_{\frac{n-1}{2}}) = \frac{1}{2}$.
> > >
> > >
> > > **2. In Equation (21) should one be taking  $\arg$ of the right-hand expression?**
> > >
> > > -  Thanks for the comment. We will include $\arg$ to the $\max$ operator in the right-hand expression in the revised paper.
> > >
> > > ----
> > > We hope that our responses can help address your concerns. If you have any other comments or feedback, please let us know! We will be happy to provide further responses.

---

### Author Response · Authors · 2021-08-06
**Response to all reviewers**

We thank all the reviewers for their careful reviews and positive assessment of our work. Below, we provide responses to each reviewer separately.  We hope that our responses can help address reviewers’ concerns. If you have any other comments or feedback, please let us know. Thank you again for the reviews!

---

### Decision · Program_Chairs · 2021-09-27

**Decision:**

Accept (Poster)

**Comment:**

Throughout the discussion the reviewers agreed unanimously that the paper provides an interesting theoretical contribution and should be accepted to NeurIPS. We urge the authors to revise the paper according to the comments provided by the reviewers in the rebuttal.

Thanks for submitting your work to NeurIPS!